# Year-round Impact of Winter Sea Ice Thickness Observations on Seasonal Forecasts

Beena Balan-Sarojini[1], Steffen Tietsche[1], Michael Mayer[1,2],

Magdalena Balmaseda[1], Hao Zuo[1], Patricia de Rosnay[1], Tim

Stockdale[1], and Frederic Vitart[1]

[1]*The European Centre for Medium-Range Weather Forecasts, , Shinfield Rd, Reading*

*RG2 9AX, UK*

[2]*Department of Meteorology and Geophysics, , University of Vienna, Vienna, Austria*

Monday 2$^{\text{nd}}$ November, 2020

## Abstract

Nowadays many seasonal forecasting centres provide dynamical predictions of sea ice. While initializing sea ice by assimilating sea ice concentration (SIC) is common, constraining initial conditions of sea ice thickness (SIT) is only in its early stages. Here, we make use of the availability of Arctic-wide winter SIT observations covering 2011-2016 to constrain SIT in the ECMWF (European Centre for Medium-Range Weather Forecasts) ocean–sea-ice analysis system with the aim of improving the initial conditions of the coupled forecasts. The impact of the improved initialization on the predictive skill of pan-Arctic sea ice for lead times of up to 7 months is investigated in a low-resolution analogue of the currently operational ECMWF seasonal forecasting system SEAS5.

By using winter SIT information merged from CS2 and SMOS (CS2SMOS: CryoSat2 Soil Moisture and Ocean Salinity), substantial changes of sea ice volume and thickness are found in the ocean–sea-ice analysis, including damping of the overly strong seasonal cycle of sea ice volume. Compared with the reference experiment, which does not use SIT information, forecasts initialized using SIT data show a reduction of the excess sea ice bias and an overall reduction of seasonal sea ice area forecast errors of up to 5% at lead months 2 to 5. Change in biases is the main forecast impact. Using the Integrated Ice Edge Error (IIEE) metric, we find significant improvement of up to 28% in the September sea ice edge forecast started from April. However, sea ice forecasts for September started in spring still exhibit a positive sea ice bias, which points to too slow melting in the forecast model. A slight degradation in skill is found in the early freezing

season sea ice forecasts initialized in July and August, which is related to degraded initial conditions during these months. Both the ocean reanalyses, with and without SIT constraint, show strong melting in the middle of the melt season compared to the forecasts. This excessive melting related to positive net surface radiation biases in the atmospheric flux forcing of the ocean reanalyses remains and consequently degrades analysed summer SIC. The impact of thickness initialization is also visible in the sea surface and near-surface temperature forecasts. While positive forecast impact is seen in near-surface temperature forecasts of early freezing season (Sep-Oct-Nov) initialized in May (when the sea ice initial conditions have been observationally constrained in the preceding winter months), negative impact is seen for the same season when initialised in August month when the sea ice initial conditions are degraded. We conclude that the strong thinning by CS2SMOS initialization mitigates or enhances seasonally dependent forecast model errors in sea ice and near-surface temperatures in all seasons.

The results indicate that the memory of SIT in the spring initial conditions lasts into autumn, influencing forecasts of the peak summer melt and early freezing seasons. Our results demonstrate the usefulness of new sea ice observational products in both data assimilation and forecasting systems, and strongly suggest that better initialization of SIT is crucial for improving seasonal sea ice forecasts.

# 1    Introduction

Sea ice is an integral part of the Earth system as it regulates the heat, moisture and momentum flux exchange between the polar oceans and the atmosphere. Decline in Arctic sea ice is a visible indicator of the changing climate. Forecasting Arctic sea ice has advanced significantly in the last decade, with most forecasting centres using prognostic sea ice models operationally, allowing us to explore the sea ice forecast skill on long lead times from weeks to months to seasons. Possibilities of economically viable shorter shipping routes across the Arctic in the summer are constantly being explored. Monthly and seasonal outlooks of sea ice products are therefore in great demand especially by the Arctic communities, maritime and resource extraction industries.

Moreover, there is increasing scientific evidence that warming and sea ice loss in the Arctic due to climate change affect the European weather and climate (Balmaseda et al. (2010), Mori et al. (2014), Overland et al. (2016), Ruggieri et al. (2016)). Unlike sea ice concentration and extent, long records of satellite observations of sea ice thickness are sorely lacking (Laxon et al. (2003), Kwok and Rothrock (2009), Haas et al. (2010), Meier et al. (2014), Sallila et al. (2019), Scarlat et al. (2020)).

Since reliable estimates of long-term, basin-wide sea ice extent and volume are needed for understanding climate change and for initializing numerical weather forecasts, there is growing interest in using improved and new types of sea ice observations in data assimilation systems (Lindsay et al. (2008), Blanchard-Wrigglesworth et al. (2011), Tietsche et al. (2013), Sigmond et al.

(2013), Balmaseda et al. (2015)). Earlier studies propose that long-term memory in the winter sea ice thickness can potentially improve summer sea ice extent forecasts (Guemas et al. (2016), Tietsche et al. (2014), Day et al. (2014)). They concluded that potential predictability mainly originates from the persistence or advection of sea ice thickness anomalies, interaction with ocean and atmosphere and changes in the radiative forcing.

While assimilation of sea ice concentration (SIC) is routinely done in operational sea ice forecasting, assimilation of sea ice thickness (SIT) is at its early stages (Allard et al. (2018), Xie et al. (2018), Mu et al. (2018), Fritzner et al. (2019)). These studies have found that SIT initialization improves sea ice forecasts in forced ocean–sea-ice forecasting systems which were run for short time periods spanning from 3 months up to 3 years. Blockley and Peterson (2018) reported for the first time the positive impact of winter SIT initialization on the skill of seasonal forecasts for summer sea ice forecasts using a fully-coupled atmosphere–ocean–sea-ice model. All of these studies used either European Space Agency's Cryosat-2 (CS2) radar altimeter freeboard SIT measurements alone (Laxon et al. (2013), Hendricks et al. (2016)) or merged with SMOS radiometric measurements (Kaleschke et al. (2012),Tian-Kunze et al. (2014)) in a dataset called CS2SMOS (Ricker et al. (2017)).

Currently SIC is the only sea ice variable assimilated in the ECMWF ocean-sea–ice data assimilation system. Although the ECMWF sea ice reanalysis and reforecasts compare well with other systems (Chevallier et al. (2017), Uotila et al. (2019), Zampieri et al. (2018), Zampieri et al. (2019)), they are affected

by noticeable errors (Tietsche et al. (2018)). There are large biases in sea ice forecasts from months to seasons, pointing to uncertainties in both the models and observations used in the assimilation and forecasting systems. Here we explore the pathway to improve the initialization using observations of sea ice thickness which covers both the thick and thin ice regions of the Arctic. We then assess the impact of the changed sea ice initial condition on the forecast skill on long lead times of months to seasons. Compared to Blockley and Peterson (2018), who looked only at summer forecast skills, our study for the first time assesses the forecast impact of SIT initialization on all seasons using a fully-coupled seasonal forecasting system. We use the ECMWF coupled ensemble seasonal forecasting system SEAS5 and CS2SMOS thickness observations.

Our study takes a forecasting system end-to-end perspective, from observations, modelling to forecast products. The rest of the article is organised as follows. Section 2 describes the methodology of sea ice thickness initialization and forecasting, including a brief description of ocean–sea-ice models, the assimilation system, the atmosphere-ocean–sea-ice coupled forecasting system, observations used and the experimental set-up. Section 3 presents the main results and has three main foci: i) assessing the impact of new SIT observations on the analysed sea ice state and the impact of the changed sea ice initialization on seasonal range sea-ice forecasts (sections 3.1 and 3.2), ii) improving Arctic sea-ice forecast skill by understanding the errors in the coupled forecast model and the data assimilation system through targeted diagnostics (sections 3.3), and iii) quantifying the impact of sea-ice improvements on seasonal forecasts of

atmospheric variables (section 3.4). Finally, Section 4 provides the summary of the findings with concluding remarks.

# 2    Observations and Methods

The procedure followed here to assess the impact of SIT information follows a twin experiment approach. Each of the experiments consists of two distinctive steps: 1) the production of a set of ocean and sea ice initial conditions by conducting twin ocean–sea-ice assimilation experiments (ocean–sea-ice reanalyses; abbreviated as ORA), which only differ in the use of SIT information ; and 2) the production of a set of twin retrospective seasonal forecast (reforecast) experiments, initialized from the respective ORA. The ORA twin reanalyses are a low resolution variant of the currently operational ORAS5 (Zuo et al. (2019)). The seasonal forecast experiments are also low resolution versions of the operational ECMWF seasonal forecasting system SEAS5 (Stockdale et al. (2018), Johnson et al. (2018)). The impact of SIT in the ocean initial conditions and seasonal forecast is then evaluated, using verification against observational datasets and other more specific diagnostics. The verification will also use fields from ORAS5 and ERA-5 (ECMWF atmospheric Re-Analysis-5); Hersbach et al. 2019) reanalyses. Although the datasets used for verification are not strictly independent, evaluation using those datasets is relevant as it allows cross-checking between variables, for instance between SIC and SIT assimilation. SIT verification using CS2SMOS dataset is also conducted as a sanity check of the nudging

approach: if the approach works, the difference with respect to CS2SMOS should be smaller in ORA-SIT than in ORA-REF. In this section we first describe the sea ice information used for both initialization and verification, and then offer a brief description of the experimental set-up.

In addition to the sea ice data sets described below, the initialization step uses ocean observations: sea surface temperature, sea-level anomalies from altimeter and in-situ temperature and salinity, which are the same as those used in ORAS5, as described in Zuo et al. (2019).

## 2.1 Sea Ice Observational Information

### 2.1.1 Sea Ice Concentration Product: OSI-401-b

The two ocean–sea-ice reanalysis experiments presented here assimilate the sea ice concentration product of the EUMETSAT Ocean and Sea Ice Satellite Application Facility (OSI SAF, www.osi-saf.org; product identifier OSI-401-b (Tonboe et al. (2017))). The Level-3 OSI SAF SIC product (OSI-401-b) is produced as daily-mean fields with only a few hours latency. In contrast to the operational ORAS5 system, which uses Level-4 SIC data, experiments presented in this study use Level-3 SIC data. The main difference is that Level-4 products rely on gap-filling, whereas Level-3 products have missing data, for instance if the satellite has a temporary malfunction, or if certain areas like the North Pole are not observed. The OSI-401-b SIC observational estimate is based on SSMIS (Special Sensor Microwave Imager / Sounder) measurements. SIC is provided as the percentage of an ocean grid point covered by sea ice. The product comes

in a polar stereographic grid of 10km horizontal resolution with varying pole hole size.

The impact of Level-3 SIC observations in the initialization is reported to have neutral forecast impact on seasonal sea ice forecasts and positive impact on sub-seasonal range (Balan-Sarojini et al. (2019)). The OSISAF OSI-401-b SIC data set is also used for verification of SIC and sea ice edge.

### 2.1.2 Sea Ice Thickness Product: CS2SMOS

A recent initiative led by the Alfred Wegener Institute provides a merged product of Arctic-wide winter ice thickness that combines thick-ice retrievals by CryoSat2 (CS2) satellite and thin-ice retrievals by the Soil Moisture and Ocean Salinity (SMOS) satellite. This merged sea ice thickness observational product, CS2SMOS (https://spaces.awi.de/display/CS2SMOS, Ricker et al. (2017)), is the first ever multi-sensor ice thickness product for the Arctic. CS2 (Hendricks et al. (2016)) measures freeboard (the height of the ice or snow surface above the water level) using altimetry, whereas SMOS (Tian-Kunze et al. (2014)) measures brightness temperatures in the L-band microwave frequencies. Both measurements are converted to ice thickness in metres. Due to their different measurement principles, SMOS retrievals should be reliable for ice thinner than about 1 m and CS2 retrievals for ice thicker than 1 m. The merged product can hence represent the entire thickness range covering the whole Arctic with reasonable accuracy (Ricker et al. (2017)). CS2 and SMOS are merged using an optimal interpolation scheme to produce the CS2SMOS product, which is avail-

able on a weekly basis on an Equal-Area Scalable Earth Grid version 2 (EASE2) grid with 25 km horizontal resolution covering all regions in the Northern Hemisphere where sea ice can be expected. Both the CS2 and SMOS retrievals are not possible in the melt season due to signal contamination owing to the presence of melt ponds, and wet and warm snow and ice surfaces, therefore it is only available for 5 full months from November to March of the ice growth season every year.

In a merged product like CS2SMOS it is difficult to appropriately represent observational uncertainties. For instance, sensor-specific errors could affect regional sea ice thickness: over multi-year thick ice in the Canadian Basin, errors associated with Cryosat-2 retrievals dominate, whereas in the Bering or Okhotsk Sea with mostly seasonal thin ice, errors associated with SMOS retrievals dominate. As reported in Ricker et al. (2017), the relative error is maximum in the thickness range of 0.5-1.0 m in the merged product, where relative uncertainty is high for both CS2 and SMOS.

The CS2SMOS SIT information without observational uncertainties has been assimilated in one of the twin ORA experiments, during the November–March period. It has also been used for verification of initialization in those months. We emphasize that this dataset does not provide SIT information during the period April–October. Nevertheless, there is still substantial impact in the April–October period from constraining sea ice thickness during the November–March period, as we will see in Section 3 – a truly year-round impact.

## 2.2 Methods

### 2.2.1 Ocean–Sea-Ice Reanalysis Experiments

In order to assess the impact of new sea ice thickness observations on the assimilation, we carry out two ORAs as shown in Table 1. They are 1) a reference experiment with SIC assimilation (ORA-REF), and 2) an experiment with SIC assimilation and sea ice thickness constraint (ORA-SIT). Experiments ORA-REF and ORA-SIT are run for the time period January 2011 to December 2016, because these are the full years for which CS2SMOS observations were available at the time of experimentation. Note that ORA-REF is a continuation of a longer experiment which started in 2005 and ORA-SIT starts from ORA-REF on the 1st of January, 2011.

| Experiment name | SIC constraint | SIT constraint | Time period | Description |
|---|---|---|---|---|
| ORA-REF | Yes | No | 2011-2016 | SIC assimilation |
| ORA-SIT | Yes | Yes | 2011-2016 | SIC assimilation and SIT nudging |

Table 1: Specifications of the ocean–sea-ice assimilation experiments.

Our reanalysis experiments are forced by near-surface air temperature, humidity and winds as well as surface radiative fluxes from the atmospheric reanalysis ERA-Interim (ERA-I) (Dee et al. (2011)) until 2015 and from the ECMWF operational analysis from 2015 to 2016. We use the same set-up of NEMOVAR

(Variational data assimilation system for NEMO (Nucleus for European Modelling of the Ocean) ocean model) used in ORAS5 (Zuo et al. (2019)) – in particular, almost the same observations are assimilated. The only differences are the following: a) a coarser model resolution as described below, b) different assimilated SIC observations compared to the current operational one and, c) a longer assimilation window of 10 days instead of 5 days.

The ocean general circulation model used in these experiments is NEMO version 3.4 (Madec (2008)) with a horizontal resolution of approximately 1° and 42 vertical layers. The grid is tripolar, with the poles over Northern Canada, Central Asia and Antarctica enabling higher resolution across the Arctic than at the equator. The first model layer is 10 m thick, and the upper 25 levels represent approximately the top 880 m. Both the horizontal and vertical resolution in our setup is lower than that of the current operational system, which has a horizontal resolution of approximately 0.25° and 75 vertical levels. The time step is one hour.

The prognostic thermodynamic-dynamic sea ice model used is LIM2 (Louvain-la-Neuve Sea Ice Model) in its original version (Fichefet and Maqueda (1997)). The vertical growth and decay of ice due to thermodynamic processes is modelled according to the three-layer (one layer for snow and two layers for ice) Semtner scheme (Semtner (1976)). The ice velocity is calculated from a momentum balance considering sea ice as a two-dimensional continuum in dynamical interaction with the atmosphere and ocean. Internal stress within the ice for different states of deformation is computed following the viscous-plastic (VP)

rheology proposed by Hibler III (1979). LIM2 has a single sea ice category to represent sub-grid scale ice thickness distribution, and open water areas like leads and polynyas are represented using ice concentration. Melt ponds are not modelled which could affect the accurate representation of surface albedo over sea-ice. However, we note that only the ocean reanalysis ORAS5 actually makes use of the albedo computed by LIM2 (which is too high in summer), while the atmospheric reanalyses used for verification and the forecasting system use the same climatological albedo (based on SHEBA campaign observations; Beesley et al. (2000)). Moreover, a recent comparison study (Pohl et al. (2020)) shows that, overall, the broadband albedo over Arctic sea-ice derived from MERIS observations is comparable to that in the ERA5 atmospheric reanalysis in terms of the seasonal cycle on large spatial scales. The forecast albedo over ice is comparable to that in ERA-5 and ERA-Interim atmospheric reanalyses. LIM2 has a time step of 1 hour and is coupled to the ocean at every time step.

As for ORAS5, both experiments here use the variational data assimilation using NEMOVAR in a 3D-Var FGAT (First Guess at Appropriate Time) configuration as described in Mogensen et al. (2012). The length of the assimilation window is 10 days in our experiments. Assimilated observations comprise temperature and salinity profiles, altimeter-derived sea level anomalies and sea ice concentration. Sea-surface temperature is constrained to observations by a strong relaxation. A global freshwater correction is added to reproduce the observed global-mean sea-level change. The assimilation of the SIC is done separately from the ocean variables, and is described in Tietsche et al. (2015) and

<sup>274</sup> Zuo et al. (2017).

<sup>275</sup>    In addition to the observations assimilated via NEMOVAR, the SIT in exper-
<sup>276</sup> iment ORA-SIT is constrained to the CS2SMOS via a linear nudging technique
<sup>277</sup> (Tietsche et al. (2013), Tang et al. (2013)). The relationship between the mod-
<sup>278</sup> elled and observed sea ice thickness in a grid point is described by the following
<sup>279</sup> equation:

$$SIT^n = SIT^m - [\frac{\Delta t}{\tau}(SIT^m - SIT^o)] \tag{1}$$

<sup>280</sup>    where $SIT^n$ is the nudged thickness, $SIT^m$ is the modelled floe thickness,
<sup>281</sup> $SIT^o$ is the observed floe thickness, $\Delta t$ is the sea ice model time step of 1
<sup>282</sup> hour, and $\tau$ is the nudging coefficient corresponding to a relaxation time scale
<sup>283</sup> of 10 days. The choice of a 10-day relaxation time scale makes sense as a
<sup>284</sup> first trial, since it is consistent with the length of the assimilation window.
<sup>285</sup> To facilitate the nudging, the CS2SMOS weekly observations in EASE2 grid
<sup>286</sup> have been interpolated to daily gridded fields in ORCA 1° grid. The weekly
<sup>287</sup> to daily interpolation is done by appropriately weighting two adjacent weekly-
<sup>288</sup> mean fields. We have also tested the sensitivity to different nudging strengths
<sup>289</sup> by running variants of ORA-SIT with a relaxation time scale of 20, 30 and 60
<sup>290</sup> days. By construction, as the relaxation time scale increases from 10 days to
<sup>291</sup> 60 days, SIT is less constrained to CS2SMOS. In this study, we only use the
<sup>292</sup> experiment with the strongest constraint (10-day relaxation time) for initializing
<sup>293</sup> the ensemble reforecasts, because this time scale fits with the length of the
<sup>294</sup> assimilation window, and we aimed for a strong observational constraint in
<sup>295</sup> order to obtain a strong forecast impact.

### 2.2.2 Coupled Reforecast Experiments

In order to assess the impact of CS2SMOS sea ice thickness initialization on sea ice forecasts, we performed 2 sets of twin coupled ocean–sea-ice-atmosphere reforecast experiments as shown in Table 2, which only differ in the ocean–sea-ice initial conditions, provided by the data assimilation experiments shown in Table 1. The reference reforecast (FC-REF) is initialized by ORA-REF, and reforecast experiment FC-SIT is initiailized by ORA-SIT. Comparison of results from these two sets of reforecasts allows quantifying the impact of SIT information on the seasonal forecasts.

| Experiment name | Start years | Lead mon | Ens. size | Initial condition | Description |
|---|---|---|---|---|---|
| FC-REF | 2011–2016 | 7 | 25 | ORA-REF | SIC initialization |
| FC-SIT | 2011–2016 | 7 | 25 | ORA-SIT | SIC and SIT initialization |

Table 2: Overview of the reforecast experiments. For each of the start years, forecasts are started on the 1st of every calendar month.

The reforecast experiments are carried out using a version of the ECMWF coupled seasonal forecasting system. The coupled model consists of the same ocean and sea ice model (NEMO3.4/LIM2) used for our reanalysis experiments, and is coupled to the ECMWF atmospheric model, Integrated Forecast System (IFS) version 43r3. It is run with a horizontal resolution of 36 km, correspond-

ing to a cubic octahedral reduced Gaussian grid at truncation TCo319 and 91 vertical levels (SEAS5 is run with IFS cycle 43r1 at the same atmospheric resolution, but with 0.25° horizontal resolution and 75 vertical levels in the ocean). The coupled model also includes the land surface model HTESSEL (Hydrology Tiled ECMWF Scheme for Surface Exchanges over Land) and the ocean surface wave model WAM. The coupling of the atmosphere and ocean is done using a Gaussian interpolation method, and the coupling frequency is 1 hour. For more details on SEAS5 see (Stockdale et al. (2018), Johnson et al. (2018)).

Both reforecasts are started from the 1st of each month of each year 2011–2016, resulting in 72 forecast start dates overall. Note that out of all months of each year in the 2011-2016 period only winter (December-April) months are directly constrained by November-March observations as the CS2SMOS data is only available for those 5 full months. The initial conditions for the remaining 7 start months (May-November) of each year are indirectly affected by the thickness constraint applied earlier in the ice growth season in the reanalysis. The forecast initialized from each start date has 25 ensemble members for both sets of reforecasts.

# 3  Results

Here we first assess the impact of sea ice thickness observations on the estimation of sea ice properties in the ORA initial conditions, and then we evaluate the impact on the skill of seasonal forecast of sea ice area, sea ice edge, sea ice

volume and 2m temperature. When possible, we use the observational datasets

for verification. However, as mentioned above, sea ice thickness and volume

(SIV) can not be verified properly for the months April-October, due to the lack

of sea ice thickness observations. In those cases, we will describe the impact in

terms of differences between experiments. We use the term pan-Arctic to refer

to all regions of the Northern Hemisphere that are potentially covered by sea

ice.

## 3.1 Impact of Sea Ice Thickness Initialization on the Sea Ice Reanalysis

Figure 1 shows the SIT bias with respect to the CS2SMOS observations for

ORA-REF (Figure 1a, c) and ORA-SIT (Figure 1b, d), for March (Figure 1a,

b) and November (Figure 1c, d). The ORA-REF suffers from large ice thickness

bias of up to 1.4 m. The predominant bias pattern is an underestimation of ice

thickness by more than 1 m in the central Arctic, and an overestimation in

the Beaufort Gyre and the Canadian Archipelago of the order of 1 m. This

pattern is present for all the months when CS2SMOS is available. In March,

widespread overestimation in the coastal Arctic seas is also present. These

biases are much reduced or absent in ORA-SIT. Most of the large-scale pattern

of underestimation and overestimation of sea ice in ORA-REF is not present

in ORA-SIT in March. However, slight underestimation over the central Arctic

and overestimation over the Canadian Archipelago still remain in November.

This is caused by the lack of SIT observations during the months preceeding

November. In contrast, the estimation of the March conditions benefit from the availability of SIT information in the preceeding winter. We note that the bias in ORA-SIT over the Laptev, East Siberian and Chukchi Seas is very small, about 0.1 to 0.05 m of magnitude (below the contour interval).

Figure 2 shows the difference in SIT between ORA-SIT and ORA-REF for March, July, September and November. The difference patterns between ORA-SIT and ORA-REF are quite consistent for all the months, characterized by a thickening of the thick ice over the Central Arctic and North of Greenland, and a thinning of the thin ice area over the Beaufort and Siberian Seas, thus enhancing the spatial gradients in the sea ice thickness distribution. The largest impact occurs in March, probably because at this month the SIT observations have been assimilated during the preceeding 5 months. The impact of SIT winter information lasts well into the summer months, with a slight clockwise displacement of the thinning, and a reduction of the thickening, which by September has roughly halved. The shift in the thinning pattern is consistent with the mean climatological transpolar Arctic drift pattern and is thus likely a consequence of the mean advection. The impact during March and November is consistent with a reduction of the bias in ORA-REF (Figure 1a and c). Since basin-scale SIT observations are not available for the end of the melt season, biases are unknown.

The thickness constraint also affects the biases in SIC. Figure 3 shows the SIC bias w.r.t. OSI-401-b SIC as well as the SIC difference between ORA-REF and ORA-SIT. In March, the month of sea ice maximum, ORA-REF shows

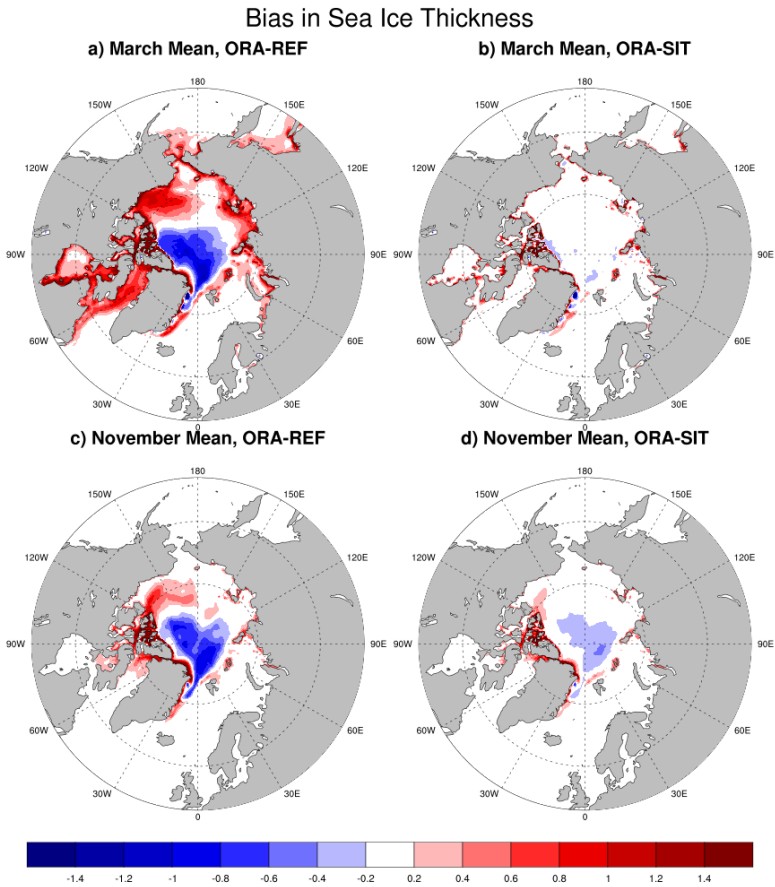

Figure 1: Bias in monthly-mean (2011-2016) sea ice thickness (m) in experiment a) ORA-REF and b) ORA-SIT, for March (a, b) and November (c, d). The reference is CS2SMOS observations. ORA-REF is the ocean–sea-ice assimilation experiment with no sea ice thickness constraint. ORA-SIT is the assimilation experiment with a thickness relaxation time scale of 10 days.

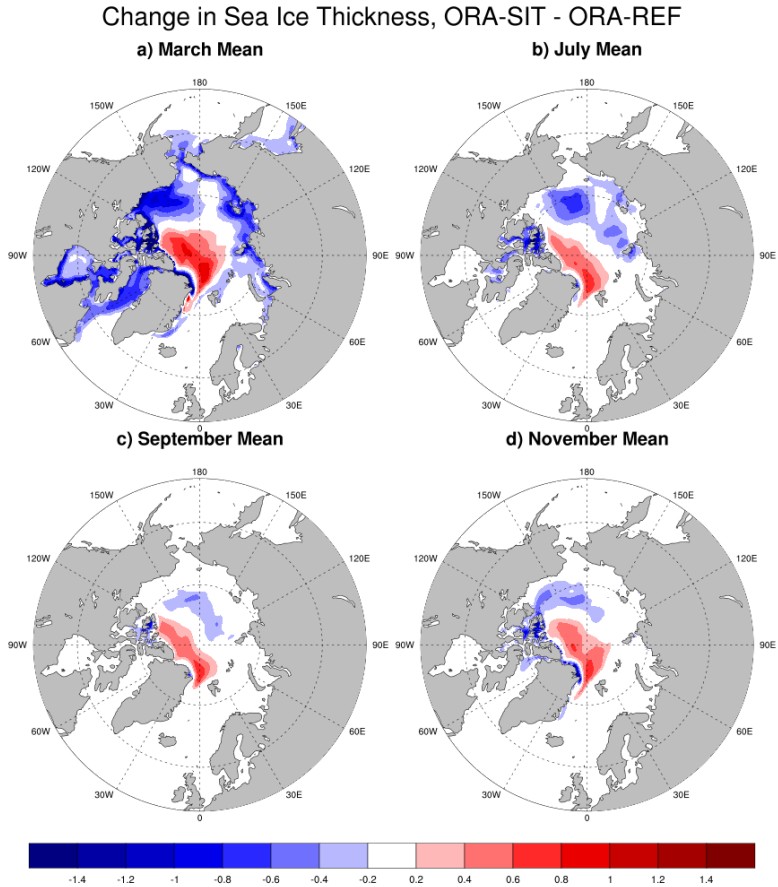

Figure 2: Difference in monthly-mean (2011-2016) sea ice thickness (m) between experiments ORA-SIT and ORA-REF for a) March and b) July and for c) September and d) November months.

**Bias and Change in Sea Ice Concentration**

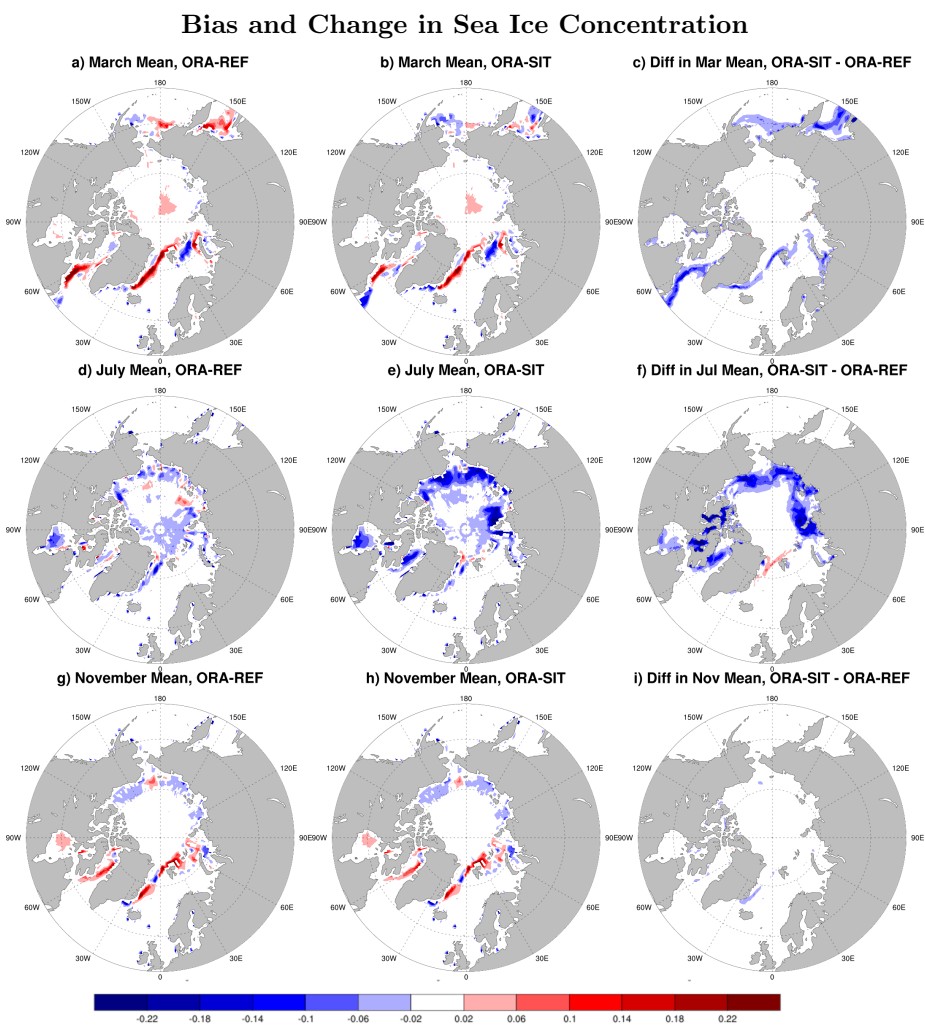

Figure 3: Bias in monthly-mean (2011-2016) sea ice concentration w.r.t. OSI-401-b observations for ORA-REF (a, d, g), ORA-SIT (b, e, h), and the difference between ORA-SIT and ORA-REF for (c, f, i). Panels (a, b, c) are for March, (d, e, f) for July, and (g, h, i) for November.

mostly an overestimation of SIC all around the sea ice edge, over the Davis Strait, northeast of Greenland, Bering Sea and Okhotsk Sea. In ORA-SIT this bias is uniformly reduced by up to 10% . In November (Figure 3g, h and i), when the sea ice edge is expanding with newly frozen ice, ORA-REF has similar SIC overestimation biases over the ice edge, but this time the SIT constraint has very little impact on SIC biases. This is because of no SIT nudging happening in the preceding months. Also, the very small changes in SIC bias between ORA-REF and ORA-SIT over Chukchi and East Siberian Sea regions of negligible ice thickness bias in ORA-SIT (Figure 1d) is suggestive of fast growth processes in the forward model which is faster than the timescales intrinsic to the SIC assimilation. The ORA-REF biases in July are characterized by a weak underestimation of SIC. Notably, in ORA-SIT there is an increase of the negative SIC bias of more than 10% over the Pacific and Siberian Arctic sectors towards the end of melt season, with July and August (not shown) months being the most affected.

To gain some insight into the degradation of the July SIC bias in ORA-SIT we look at the mean annual cycle of the SIC assimilation increments. The assimilation increments are indicative of the corrections that the assimilation of SIC observations exerts to compensate for errors in the sea ice state. Figure 4 shows the mean annual cycle of the area-averaged assimilation increments in ORA-REF (blue) and ORA-SIT (green). In both experiments, the assimilation increments exhibit a clear seasonal cycle, with large positive increments from May to October, indicative of strong underestimation of SIC in the forward

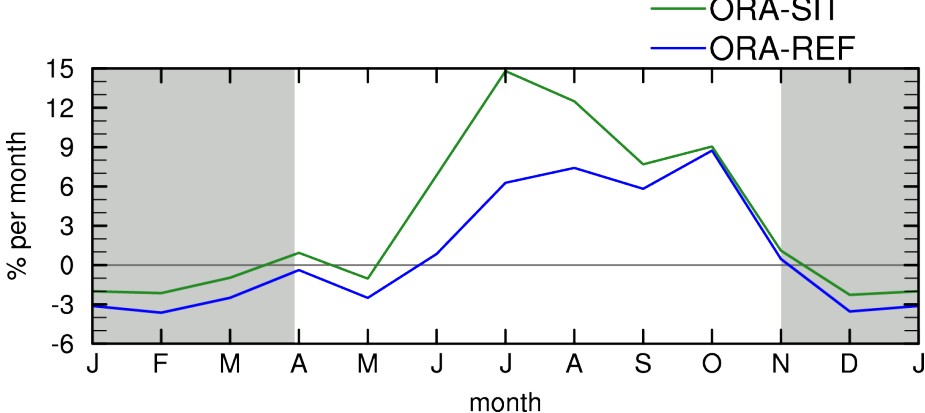

Figure 4: Annual cycle of the mean of the SIC increments in ORA-SIT (green), and ORA-REF (blue), averaged over north of $70°N$ during 2011-2016. The grey shading shows months (January to March, and November to December) with CS2SMOS SIT nudging.

model, and weak negative increments from December to March. The differences in SIC increments over the Arctic between the two experiments peaks during July, with ORA-SIT increments about 9% per month higher than in ORA-REF. The results in this figure indicate that 1) both ORAs melt sea ice too fast during the summer months, as shown by negative SIC biases in the marginal seas of the Arctic Ocean where thin sea ice resides during the summer months (Figure 3d and e) ; and 2) the SIT assimilation exacerbates the summer SIC biases in ORA-SIT (as seen in eg: Figure 3e) due to corrected but thinner sea ice at the begining of the melt season in almost all marginal seas of the Arctic Ocean (Figure 2a).

Figure 5: Bias in the forecast of pan-Arctic sea ice area ($\times 10^6 \text{km}^2$) w.r.t. ORAS5 as a function of start and lead month for 2011–2016, a) in the reference reforecast FC-REF and b) in the SIT-initialised reforecast FC-SIT. Red colour denotes over-prediction of sea ice area, and blue colour denotes under-prediction.

From January to May and from November to December, on an average less ice is being taken away by the increments in the ORA-SIT (green) analysis than that in ORA-REF (Figure 4). These results clearly show the long-lasting effect of the SIT information: the SIT constraint was only applied during the growth season from November to March (grey shading), but its impact,whether positive or negative, is evident in sea ice concentration throughout the melting season even in the presence of SIC assimilation.

## 3.2 Impact of Ice Thickness Initialization on Sea Ice Forecasts

Figure 5a gives an overview of bias in sea ice area in the FC-REF reforecast w.r.t. ORAS5 reanalysis as a function of forecast start and lead months. ORAS5 is preferred to OSISAF for the verification of integrated sea ice area because of its complete spatial coverage. The figure shows the forecast bias for different forecast lead times (y-axis) as a function of forecast starting month (x-axis). Errors at lead month 1 are generally small throughout the year. However, for longer lead times, there is a strong over-prediction of sea ice area in summer months, and a moderate under-prediction of autumn sea ice conditions, consistent with too slow melting and refreeze respectively. The forecast biases are generally small in winter months.

These three bias regimes, in general – small bias in winter, positive bias in summer and negative bias in autumn – seem to be mostly independent of start months. These biases shown in FC-REF are quite similar to those in SEAS5 (not shown) which are discussed in more detail in Stockdale et al. (2018). The positive biases in the melt season forecasts are considerably reduced with the SIT initialisation in FC-SIT started in January to June and the negative biases in the forecasts is worsened in FC-SIT started in July to October (Figure 5b). The forecasts for winter months remain unbiased in FC-SIT. Note that the bias regimes in the forecasts are very different from the bias regimes in the reanalysis (Section 3.1), which tends to have too much ice in winter and too little ice in summer.

⁴³⁹ Impact of thickness initialization has not only improved the biases in summer

⁴⁴⁰ SIC forecasts, but it has also improved the sea ice edge forecasts as measured by

⁴⁴¹ the Integrated Ice Edge Error (IIEE) (Figure 6). Seasonal forecasts of ice edge

⁴⁴² are in great demand for exploring economically viable Arctic shipping routes.

⁴⁴³ IIEE is one of the recent user-relevant sea ice metrics on ice extent or ice edge

⁴⁴⁴ (Goessling et al. (2016), Bunzel et al. (2017)). Since it can be decomposed into

⁴⁴⁵ over- and under-prediction, it is more useful than the traditional basin-wide sea

⁴⁴⁶ ice extent error.

⁴⁴⁷ For simplicity, we assess ice edge forecasts by using the deterministic IIEE

⁴⁴⁸ metric calculated from the ice edge of the ensemble mean SIC forecasts. We have

⁴⁴⁹ also tested probabilistic metrics like the Spatial Probability Score suggested by

⁴⁵⁰ Goessling and Jung (2018) and found that they give very similar results.

⁴⁵¹ IIEE for all lead months and start months verified against OSI-401-b sug-

⁴⁵² gests reduced error in sea ice edge (blue colours) in FC-SIT overall. The most

⁴⁵³ striking feature is the significant improvement in summer forecast error for lead

⁴⁵⁴ months 2–7 in FC-SIT compared to FC-REF. The main contribution to the er-

⁴⁵⁵ ror reduction of up to 30% in summer forecasts comes from the reduction of the

⁴⁵⁶ model bias leading to consistent over-prediction (not shown). For the traditional

⁴⁵⁷ September sea ice extent forecast starting in April, an improvement of 28% is

⁴⁵⁸ found. Forecast verification in October and November from July and August

⁴⁵⁹ starts show a slight degradation, caused by under-prediction (not shown). This

⁴⁶⁰ could again be due to the indirect effect of a thinner starting point in FC-SIT

⁴⁶¹ (Figure 2b) and a lower, degraded SIC in the ORA-SIT reanalysis (Figure 3e),

**Difference in Integrated Ice Edge Error**

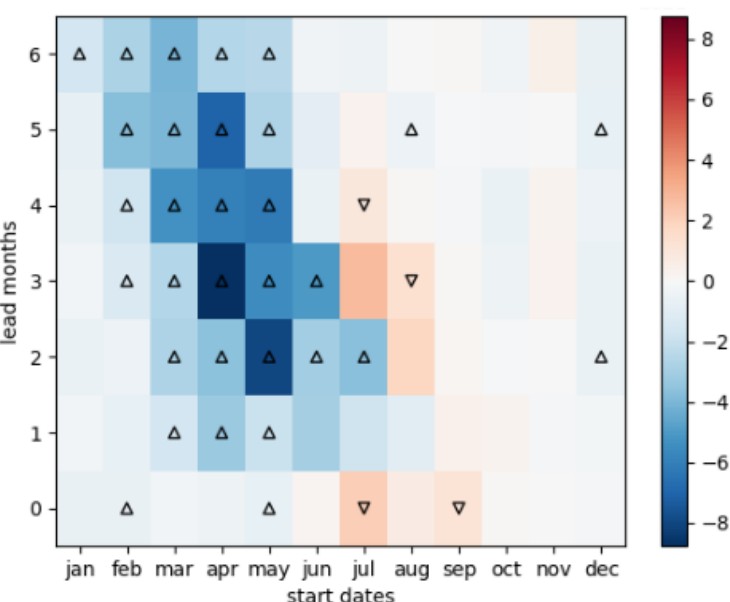

Figure 6: Difference in Integrated Ice Edge Error in $10^5 \, \mathrm{km}^2$ between FC-SIT and FC-REF reforecasts 2011–2016 w.r.t. OSI-401-b observations. Blue colour denotes reduced error in sea ice edge in FC-SIT and red colour denotes increased error in FC-SIT. Black triangles represent statistical significance at the 5% level according to the sign test (DelSole and Tippett, 2016)

.

**Mean Absolute Error in SIC Forecasts**

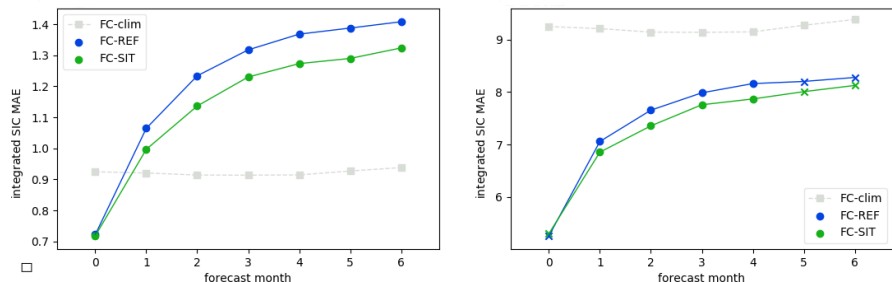

Figure 7: Spatially integrated SIC mean absolute error over lead month for all FC-REF and FC-SIT forecasts (72 forecasts each first of the month from January 2011 to December 2016) w.r.t OSI-401-b observations. Panel a) shows the error in $10^6\,\mathrm{km}^2$ without bias correction, panel b) the error in $10^5\,\mathrm{km}^2$ after bias correction. Lead months for which the reduction of forecast error in FC-SIT passes a statistical significance test at the 5% level ((DelSole and Tippett, 2016)) are marked by filled circles, insignificant changes are marked by crosses. The error of a simple climatological reference forecast is also shown as FC-clim.

<sub>462</sub> combined with the already existing slow refreeze nature of the model.

<sub>463</sub> Figures 5 and 6 point out that the impact of ice thickness initialization on
<sub>464</sub> the forecast bias and errors is strongly dependent on season and lead time. Most
<sub>465</sub> seasons and lead times are improved but some are, perhaps inevitably, deterio-
<sub>466</sub> rated. To measure the overall impact on forecast error and make a statement
<sub>467</sub> about potential skill improvements that are to be expected for operational fore-
<sub>468</sub> casts, we aggregate FC-SIT and FC-REF for all start months from January
<sub>469</sub> 2011 to December 2016 and compute the area-integrated mean absolute fore-
<sub>470</sub> cast error (MAE) of sea ice concentration for each lead month. In order to

obtain the bias-corrected forecast value, for each combination of grid cell, start date and forecast lead time, we compute the mean forecast error over all forecasts, and then subtract it from the "raw" forecast value. Comparison against a climatological benchmark forecast is a very useful background information for evaluating the predictive skill of ensemble forecasting systems (e.g. Zampieri et al. (2018)). The climatological reference forecast for a given target month and year is constructed by using the verification data valid for the same calendar month but different years from the range of target months considered (January 2011 to June 2017).

Averaged over all start dates and grid points, Figure 7 shows that the MAE of sea ice area is substantially improved in FC-SIT. When no bias correction is applied prior to computing the MAE (Figure 7a), FC-SIT forecasts are significantly better in each lead month, with maximum error reduction of about 10%.

However, skill assessments of seasonal forecasts are conventionally made after a forecast calibration where the impact of the forecast bias is removed. By this measure, a reduction of forecast bias does not by itself count as an improvement. As Figure 7b shows, removing the respective bias of FC-SIT and FC-REF prior to computing the MAE results in a smaller error reduction: errors in FC-SIT are significantly lower only in lead months 2–5, by up to 5%. Figure 7 demonstrates that, although the thickness initialization predominantly reduces the bias, it also leads to an improvement in the skill of sea ice area forecasts that is relevant for operational forecasting systems.

The importance of forecast biases is illustrated by benchmarking the errors of the dynamical forecasting system against a simple statistical reference forecast: Figure 7 also shows the errors of a climatological reference forecast (FC-clim). Without bias correction, errors of both FC-REF and FC-SIT are larger than those from FC-clim already after one month, while after bias correction, both FC-REF and FC-SIT have lower errors than FC-clim for all lead months.

Finally, we analyse the impact of SIT initialization on forecasts of pan-Arctic sea ice volume. Though an integrated quantity like pan-Arctic sea ice volume is a result of many dynamic and thermodynamic sea-ice processes and lacks regional details, it is a key indicator for understanding of the Arctic energy cycle, an important process that needs to be realistically represented in reanalyses and seasonal forecasts. It is useful to compare the contrasting SIV seasonal cycles in coupled and uncoupled mode, and with/without SIT observational constraint in the initialization, since this helps to identify the origin of errors in the systems in the specific operational set up. Figure 8 shows the sea ice volume forecast climate at different lead month for the forecasts starting in May (top) and August (bottom). The forecast climate is computed by averaging the reforecast starting at a given calendar month for the years 2011-2015. Seven months forecasts started in August lead to February of the following year. Since the ORAs are not available in January and February, 2017, the year 2016 is not accounted for in this figure. For reference, the sea ice volume estimates of ORA-REF and ORA-SIT reanalyses are also shown. It is remarkable that the shape of the seasonal cycle is largely preserved between FC-REF and FC-SIT,

**Time Evolution of Mean Sea Ice Volume Forecasts**

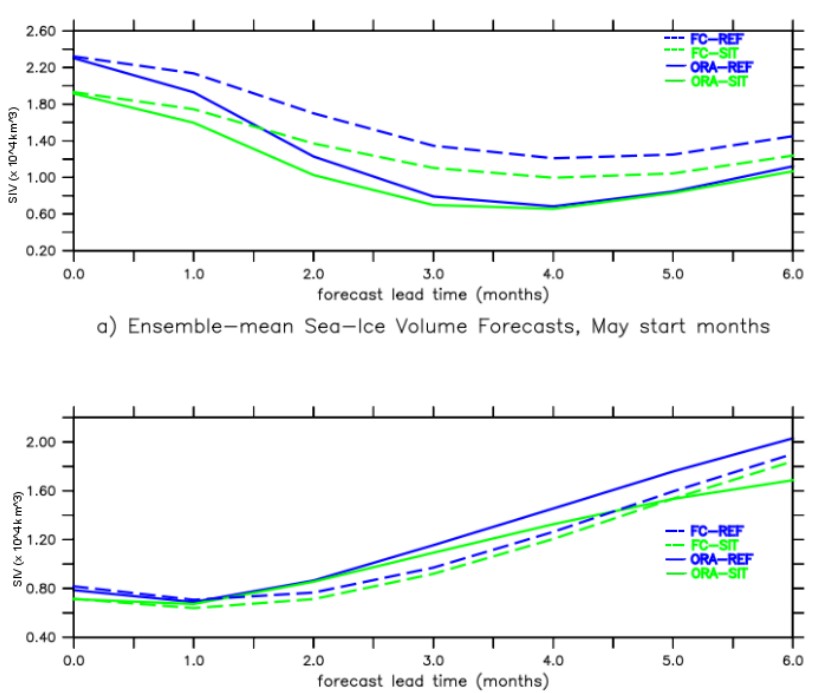

a) Ensemble—mean Sea—Ice Volume Forecasts, May start months

b) Ensemble—mean Sea—Ice Volume Forecasts, August start months

Figure 8: Time series of ensemble-mean sea ice volume (units are $10^4$ km$^3$) forecasts averaged over 2011–2015, for May start date (a) and August start date (b) in reference reforecast (FC-REF, dashed blue line) and reforecast with thickness initialization (FC-SIT, dashed green line) compared to their own re-analyses, ORA-REF (solid blue line), and ORA-SIT (solid green line).

maintaining the initial offset during the whole forecast range. The figure clearly shows that FC-SIT starts from a thinner ice state than FC-REF in both initial months.

The May starts show large differences between the forecasts and the ORAs: Both FC-SIT and FC-REF show a slower SIV decrease (lower melt rate) than the ORAs from June to September, and also a slower refreeze during October and November. The explanation for the different behavior of the ORAs and the forecasts is that the ORAs are constrained by the same SIC (but not the same SIT) information in summer, which leads to the convergence of the sea ice state in the ORAs during that time of the year (also seen in Figure 4). In the coupled forecasts, there is no similar constraint and they tend to converge slower than the ORAs. The melt rate of the ORAs here are consistent with those in ORAS5 (see Uotila et al. (2019) or Mayer et al. (2019)). Compared to the May starts, differences between FC-SIT and FC-REF are smaller in the August starts, and so is their agreement with the ORAs. Again, the FC-SIT shows smaller values than FC-REF from the begining, and both forecast sets exhibit a parallel SIV evolution. The shape of the seasonal cycle in the forecasts is different from the ORAs; the forecasts initialized in August show a slower refreeze during October than the ORAs. However, after October, the SIV increases faster in the forecasts than in ORA-SIT, and it continues increasing more or less at the same rate until the end of January in the forecasts, while in ORA-SIT (solid green line) the freezing rate slows down after November. As a result by the end of January the forecast SIV is higher than in ORA-SIT. ORA-REF without the

 thickness constraint has the highest SIV in the ice growth season. In the next

 section we examine the discrepancies in SIV changes between ORAs and FCs

 in more detail.

## 3.3 Linking Sea Ice Analysis and Forecast Errors to the Arctic Surface Energy Budget

In order to investigate the physical causes of sea ice errors in the ORAs and

forecasts, the Arctic surface energy budget is considered. We estimate melt

energy tendency (MET), which is the energy used to melt sea ice and energy

released in the process of freezing, and is proportional to SIV changes. It is

defined as in Mayer et al. (2019):

$$MET = L_f \rho \left(\frac{\partial SIT}{\partial t}\right) \tag{2}$$

where $L_f$ denotes latent heat of fusion (-0.3337x$10^6$ J kg$^{-1}$), $\rho$ represents

sea ice density (assumed constant at 928 kg m$^{-3}$), and $SIT$, the grid-point

averaged sea ice thickness. Thickness changes are computed as exact monthly

differences. MET can also change dynamically through lateral ice transports,

but here we average over the ocean area north of $70°N$, which should be a

sufficiently large area to average out any dynamical effects and should mainly

leave thermodynamic effects as the drivers of MET. Figure 9 shows the MET

mean annual cycle (2011-2015) north of $70°N$ for ORA-REF, ORA-SIT, FC-

REF, and FC-SIT. In order to isolate the changes in MET when switching from

forced ORA mode to coupled forecast mode and to avoid seeing mainly the effect

of feedbacks arising from the model sea ice state drifting away from the analyzed state (most notably the ice-albedo feedback), we compile the annual cycle of forecasted MET from lead-month 1 data from each start date. Assimilation increments of SIC proportionally affect SIV in the ORAs (Tietsche et al. (2013), Tietsche et al. (2015)). The resulting MET increments are shown for both ORA-REF and ORA-SIT as well. We note that the MET annual cycle of ORA-REF is very similar to that of ORAS5 (not shown) and that the average of the MET annual cycle in the ORAs is close to zero (in fact about $+0.3$ W/m$^2$ (Mayer et al. (2016), Mayer et al. (2019)), in agreement with the long-term sea ice melt), while it is $-4.8$ W/m$^2$ in FC-REF.

Figure 9 clearly shows that ORA-REF exhibits the most pronounced annual cycle of MET, with strongest melting in summer and strongest freezing in winter. Earlier studies have shown that the MET annual cycle is exaggerated in ORAS5 (Uotila et al. 2019; Mayer et al. 2019) and hence also in ORA-REF. ORA-SIT has a damped MET annual cycle, as the thickness constraint during winter prevents overly strong SIV accumulation. Lower SIV at the end of winter consequently leads to weaker melting in summer. However, summer melt in ORA-SIT is likely still too strong, as this experiment features a negative SIC bias in summer despite realistic SIT and SIC earlier in the year, when CS2SMOS data is available (see Figure 3e).

Both FC-REF and FC-SIT agree very well with each other and exhibit a much weaker MET annual cycle than the ORAs (Figure 9). The difference between the forecasts and the ORAs in May and June melting cannot be ex-

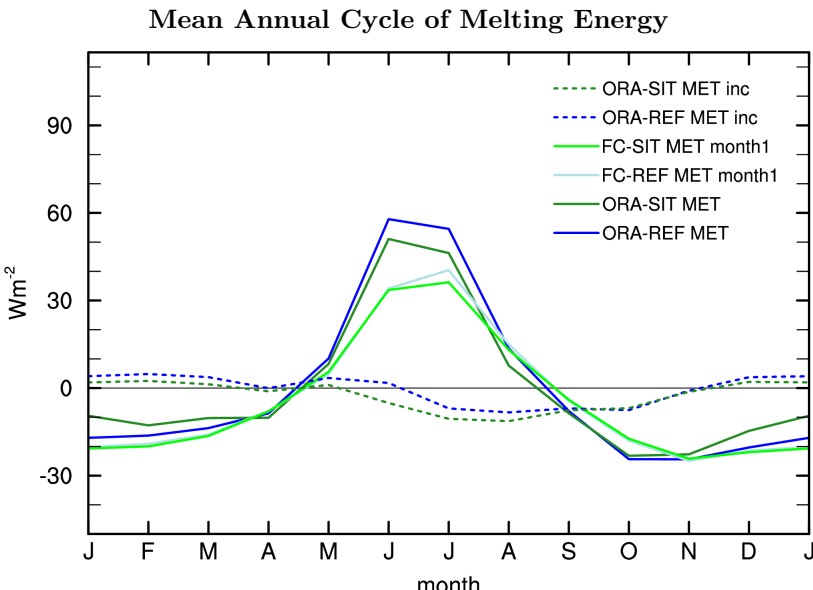

Figure 9: Mean annual cycle of MET over ocean area north of $70°N$ in ORA-REF, ORA-SIT, FC-REF (lead month 1), FC-SIT (lead month 1). MET increments for ORA-REF and ORA-SIT are shown as well.

Figure 10: a) Mean annual cycle of surface net radiation, $Rad_S$ (W/m$^2$) over ocean area north of $70°N$ from ERA-I, ERA5, FC-REF (lead month 1), FC-SIT (lead month 1), and CERES-EBAF, and b) Mean deviation of $Rad_S$ from CERES-EBAF for FC-REF, FC-SIT, ERA-I and ERA5.

plained by the MET increments (neutral impact at that time), which points to differences in energy fluxes into the sea ice as a cause.

We therefore compare the mean annual cycle of surface net radiation ($Rad_S$) over ocean north of $70°N$. Figure 10a shows the 2011-2015 annual cycle of $Rad_S$ from FC-REF, FC-SIT, ERA-I, ERA5, and the satellite-based product Clouds and Earth's Radiant System – Energy-Balanced and Filled Surface edition 4.0 (CERES-EBAF; Kato et al. (2018)), which we use as reference.

We consider $Rad_S$ from ERA-I as a good proxy for $Rad_S$ seen by the ORAs, due to two reasons: 1) ORAs use ERA-I forcing during most of the study period, and 2) ORAs does not output $Rad_S$ term; although it is not exactly identical e.g. due to different albedo in the ORAs. ERA-I exhibits a positive $Rad_S$ bias in summer, peaking in June at  15 W/m$^2$, while FC-REF and FC-SIT agree quite well with CERES-EBAF, especially in May and June, when MET discrepancies with the ORAs are large (Figure 9). Thus the $Rad_S$ bias of ERA-I can explain a large fraction of the overly strong MET in the ORAs during May and June, and the discrepancy between the ORAs and the forecasts.

The mean deviation of $Rad_S$ from CERES-EBAF (Figure 10b) clearly indicates that forecasts are closer to the observational product than the atmospheric reanalyses in May and June. This positive $Rad_S$ bias of ERA-I should be considered alongside the results by Hogan et al. (2017), who found a negative bias in downwelling shortwave radiation in ERA-I due to excessive low-level clouds. Our results can be explained by the positive bias in downwelling longwave radiation in ERA-I outweighing the shortwave flux bias. Figure 10 also shows

results for ERA5, which is closer to CERES-EBAF than ERA-I, which indi-
cates a reduced cloud bias in this more recent atmospheric reanalysis and gives
rise to the expectation of improved MET in future ocean reanalyses forced by
this product. We also note that the mean difference in sensible heat fluxes in
ERA-Interim and the forecasts and differences over sea ice were uniformly small
(generally $<2$ W/m$^2$ in summer; not shown), confirming that differences in this
field cannot explain the found differences in MET.

Additional information on the realism of summer MET in the forecasts can
be obtained from the sea ice area forecast bias of FC-SIT, as displayed in Fig-
ure 5b. It shows that FC-SIT May starts exhibit a strongly reduced positive bias
compared to FC-REF. The bias reduction can be attributed to the improved
initial conditions in FC-SIT, but the fact that the sea ice area bias remains
positive from July onward indicates that MET in the forecasts is too low in
summer. Figure 10b suggests that $Rad_S$ is too low in the forecasts in July
and August, which probably contributes to the positive SIA bias remaining in
FC-SIT (Figure 5b).

The October MET (Figure 9) indicates stronger refreeze in the ORAs (lower
MET values) compared to the forecasts. This is consistent with negative MET
increments present in the ORAs, which act to speed up refreeze in the reanalyses
(see Figure 9). The less negative MET values of the forecasts in October are
consistent with the too weak freezing and consequent underestimation of sea ice
in autumn in the August starts.

Area-averaged net radiation of all considered products agrees well with

CERES-EBAF in October (see Figure 10), and also difference maps show only a weakly positive $Rad_S$ bias of the reanalyses and forecasts compared to CERES-EBAF (not shown). Hence, errors in other physical terms such as ocean-ice fluxes must play an important role in fall, but more detailed investigations are beyond the scope of this paper.

## 3.4 Impact of Ice Thickness Initialization on Forecasts of Atmospheric Variables

To discuss the impact of the sea ice thickness constraint on the atmosphere, we first assess the differences in the forecast means (or biases) between FC-SIT and FC-REF. Figure 11a shows the bias in 2m temperature (t2m) (averaged over $50 - 90°N$) in FC-REF as a function of start dates and lead months. When verified against ERA5, significant cold biases are present in forecasts for most of the start months and lead months except for non-significant warm biases in November forecasts started in August, September and October months. We note that using atmospheric or ocean reanalysis without realistic representation of snow over sea ice, and sea ice thickness, for the verification of pan-Arctic surface temperature can be misleading, since there is large uncertainty associated with these products (Batrak and Müller (2019)). Verifying against observations is not easy, since due to the scarcity of observational campaigns over sea ice, the verification will have large representativeness error, and hence is not suitable for seasonal forecasts verification. Mean differences in t2m (Figure 11b) are generally positive with very few and non-significant exceptions, which is expected

**Difference in Mean T2m and Mean Sea Level Pressure Forecasts**

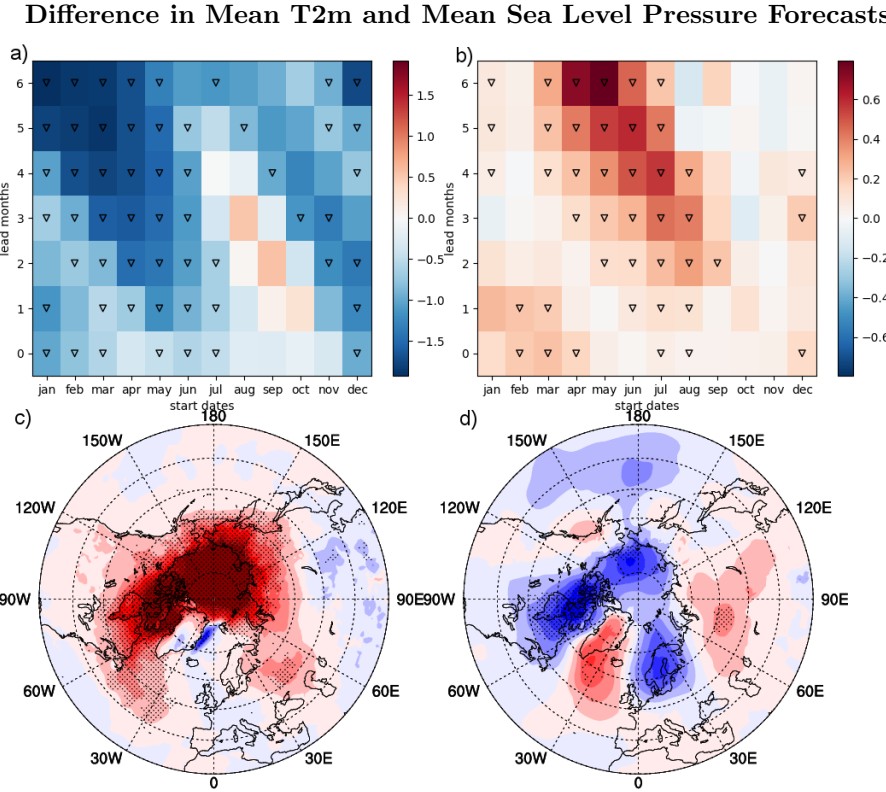

Figure 11: Mean forecast differences between FC-SIT and FC-REF 2011-2016: a) bias in mean 2m temperature (in K) north of $50°N$ w.r.t. ERA5, as a function of start dates and lead months, in FC-REF, b) similar to a), but difference in mean 2m temperature (in K) between FC-SIT and FC-REF. Triangles denote significant changes according to the sign test as recommended by DelSole and Tippett (2016) at the 5% level. Mean forecast difference (FC-SIT - FC-REF) for SON aggregated from May, June, July, August start dates of c) 2m temperature and d) mean sea level pressure. Dots indicate areas of significant changes on the 95% level according to Komolgorov-Smirnov test.

from the generally reduced sea ice cover in FC-SIT. Strongest warming with area averages of 0.5K can be found during fall for forecasts started between March and September. February and March start dates show a moderate but significant warming at short lead times, but otherwise changes are relatively small for October to February start dates. Also, changes in summer temperatures are small compared to those in fall. Inspection of temperature difference patterns between FC-SIT and FC-REF indicates that differences in summer are confined to areas around the sea ice edge (not shown), while changes in fall are more widespread (see Figure 11c). The warming pattern in fall appears as a diagonal feature in Figure 11b, which suggests that changes depend more on season than on forecast lead time. Therefore, to gain more insight into the spatial structure of the changes, Figure 11c and d show forecast differences in 2m temperature and mean sea level pressure in SON, respectively. To find robust changes, the differences are aggregated from forecasts started between May and August, yielding samples of 600 forecasts. Moreover, aggregation along the diagonal maximizes the signal (compare to Figure 11b).

Widespread temperature differences >1K can be seen over the Arctic Ocean and the Canadian Achipelago in SON (Figure 11c), but significant warming spreads also south to North America and Eurasia. Solar radiation in the Arctic is very weak for SON. Hence, the warming in FC-SIT must stem from enhanced fluxes of heat from the ocean to the atmosphere, with a possible positive feedback from increased atmospheric water vapour. The fluxes are enhanced in FC-SIT due to larger areas of open waters and increased SSTs, both a result of

reduced sea ice concentration. Furthermore, we find warming over the Northwest Atlantic, which is related to the warmer SSTs present already in the initial conditions from ORA-SIT (not shown). Another area of significant warming in FC-SIT relative to FC-REF can be seen over Eastern Europe and Western Russia. This warming seems consistent with patterns of mean sea level pressure differences shown in Figure 11d. They show lower pressure in FC-SIT over Scandinavia and higher pressure over central Russia, which together suggest more southerly winds in the region of warmer temperatures. Furthermore, mean sea level pressure changes indicate lower pressure over the Arctic Ocean and the Canadian Archipelago, i.e. in areas of maximum warming. In addition, there are positive pressure differences southeast of Greenland. Altogether, the patterns in sea level pressure difference resemble a wave-like response, but it should be kept in mind that only some parts of these changes are statistically significant. Nevertheless, we note that qualitatively similar and significant changes are also found in 500hPa geopotential forecasts for SON (not shown), suggesting that the features seen in Figure 11d are indeed robust.

We now turn to the question whether changes in the forecast mean constitute a forecast improvement or a forecast deterioration in the sense that they lead to an overall reduction of model biases. Since forecast bias is strongly dependent on region, season and lead time, aggregating over many seasons and lead months hampers physical understanding of the impact of thickness initialization. We therefore focus only on forecasts for the September–November (SON) season, where the impact on 2m temperature is strongest.

**Bias and Difference in MAE in T2m Forecasts**

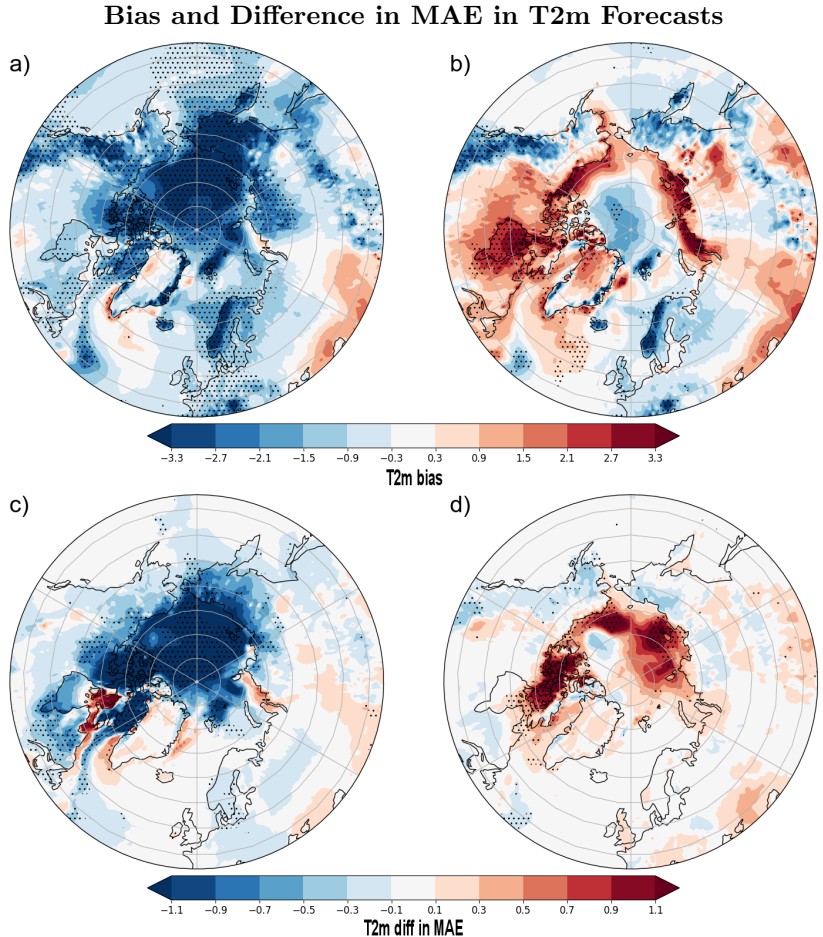

Figure 12: Bias and difference in MAE of 2m temperature against ERA5 for SON forecasts started in May (a,c) and August (b,d) respectively: Bias (in K) of FC-REF is shown on the top (a,b), and MAE difference (in K) between FC-SIT and FC-REF at the bottom (c,d). Differences significant at the 5% level according to the sign test as recommended by DelSole and Tippett (2016) are stippled. The homogeneous warming of FC-SIT w.r.t. FC-REF shown in Figure 11c results in MAE for SON t2m being reduced for May start dates c) and increased for August start dates d).

As Figure 12a and b show, the 2m-temperature forecast bias for the SON season have a strong dependence on the start and lead month. Cold biases are clearly dominating the entire hemisphere in May forecasts, whereas a mixture of warm and cold biases is visible in August forecasts, with predominantly warm biases over the ice edge. As discussed previously, the thickness initialization leads to a homogeneous warming of 2m temperature (Figure 11c), which is not very sensitive to the time of initialization.

To determine whether the mean change leads to an increase or a reduction in the bias, we compute changes in mean absolute error (MAE) of 2m-temperature forecasts without the usual calibration. This is shown in Figure 12c and d. Mean absolute forecast errors are substantially reduced in SON (by more than 1K) over the entire ice cover and some adjacent regions (Figure 12c). In this case, the thickness initialization helps to mitigate the model bias. Conversely, when initializing forecasts in August, mean absolute forecast errors are increased over the marginal Seas of the Arctic Ocean and the Canadian Archipelago (Figure 12d). This points to an exacerbation of the model biases by the thickness initialization. However, the negative impact for August start dates is not as significant as the positive impact for May start dates.

Forecast skill changes on other atmospheric fields have been explored as well. The picture for circulation-related fields such as mean sea-level pressure and 500 hPa, geopotential height (not shown) is less clear compared to 2m-temperature, indicating that much of the statistically significant changes found at the near-surface temperature in the Arctic are due to local thermodynamic

effects.

# 4  Summary and Concluding Remarks

In this paper we use 6 years of Arctic-wide sea ice thickness observations of January, February, March, November and December months during 2011 to 2016 to constrain the modelled sea ice thickness, and study the impact on the ocean–sea-ice reanalysis. Coupled forecasts of the ocean–sea-ice-wave-land-atmosphere are initialized using these data assimilation experiments, and the forecast skill of pan-Arctic sea ice for lead times up to 7 months is investigated. To our knowledge this study provides the first comprehensive assessment of coupled seasonal sea ice forecasts with thickness initialization for all the seasons. It complements to the study by Blockley and Peterson (2018), who reported the positive forecast impact on summer season only. This paper does not delve into the technical implementation of sea ice observational information in the ECMWF systems as reported in Balan-Sarojini et al. (2019), but instead it focuses on 1) collating the relevant scientific results on the impact of sea ice thickness information alone on seasonal forecasts, 2) conducting targeted diagnostics to gain understanding of the results, and 3) providing a more thorough discussion on the impact.

Constraining initial conditions by nudging to CS2SMOS ice thickness results in a substantial reduction of sea ice volume and thickness in the ocean–sea-ice analysis. This reduces some of the existing forecast biases in SEAS5 and improves forecast skill in the melt season, but in turn increases the errors during

autumn, when the existing sea ice forecast bias is negative.

The impact of sea ice thickness initialization on seasonal forecast skill for Arctic sea ice variables, namely sea ice cover, sea ice thickness, sea ice volume and sea ice edge, is mostly positive for seasonal forecasts started from January to June start dates. We find significant improvement of up to 28% in the traditional September sea ice edge forecasts started from April start dates as shown by Integrated Ice Edge Error. However, sea ice forecasts for September started in spring still exhibit a positive sea ice bias, which points to too slow melting in the forecast model. Neutral forecast impact for November and December start dates is found. However, a slight degradation is seen in autumn forecasts started from July and August start dates, which is shown to be due to errors in the sea ice initial conditions. Both the ocean reanalyses, with and without SIT constraint, show strong melting in the middle of the melt season compared to the forecasts. This excessive melting is shown to be due to positive net surface radiation biases in the atmospheric flux forcings of the ocean reanalyses. Compared to the forecasts, strong freezing is seen throughout the freeze season in the ocean reanalysis without SIT constraint. With SIT constraint applied from November to March, the existing strong freezing is somewhat damped in the late freeze season. The exact causes of the differences in freezing between the reanalyses and forecasts require further investigation. Aggregating all the forecasts started in January to December, positive forecast impact of up to 5% skill improvement for integrated SIC is found at 2-5 lead months. Thinning of sea ice by CS2SMOS mitigates or enhances seasonally dependent forecast model

error.

We reiterate that the sea-ice thickness observations are only available and assimilated for November-March. The ORA-SIT sea ice thickness from April-October is not constrained by observations. The fact that ORA-SIT has larger errors than ORA-REF in SIC for July is attributed to the overestimation of the melt in the forced model. The negative summer SIC bias gets worse in ORA-SIT than that in ORA-REF due to the fact that the ORA-SIT starts from a thinner ice state compared to ORA-REF without CS2SMOS thinning. Indeed, the assimilation of sea-ice concentration is trying hard to compensate for this excess of sea-ice melt as seen in the annual cycle of the pan-Arctic sea ice increments and melting energy tendencies. The reasons for this excess sea-ice melt during the summer season is investigated and is attributed to errors in forcing fluxes in the ORAs as just summarised. This key result points out that the dominant source of error lies in the atmospheric forcing rather than in the sea-ice model formulation or data assimilation in our experiments, and indicates that improved atmospheric fluxes from atmospheric reanalyses is urgently needed to improve the Arctic sea-ice related forecasts.

In this work we have only taken the very first step in SIT assimilation by using a simple nudging method to constrain SIT without considering the observational uncertainties. An area which needs to be explored in future studies of SIT assimilation is the use of thickness uncertainities. For instance, the uncertainty in the retrievals could be taken into account by perturbing the observations in the ensemble of data assimilations. We also note that this study does not cover

recent sea-ice model improvements such as modelling sea-ice processes affecting the sea-ice melt/growth, which are being considered for inclusion in upcoming versions of the ECMWF forecasting systems. The use of multi-category sea ice models in coupled forecasting systems is another step forward in this direction. Since uncertainty of Arctic seasonal sea ice forecasts is reported to grow at a higher rate over thin ice regions than over the central Arctic (e.g. Blanchard-Wrigglesworth et al. (2017)), we recommend observational constraint of SIT for both the thick and thin ice regions in ORAs.

The impact of sea ice thickness initialization on atmospheric variables has also been investigated. Changes in ensemble mean 2m-temperature over the pan-Arctic region are significant for SON forecasts initialized from May to August start dates. The impact is also seen in mean sea level pressure and to certain extent in 500hPa geopotential height and is mostly local and thermodynamically driven, except for some remote impact over the north west Atlantic ocean. Similar to the sea ice edge forecasts, positive forecast impact is seen for 2m-temperature forecasts for the early freezing season, SON, started in May and negative impact for the same season is seen when started in August when the initial conditions are degraded. Statistically significant changes in 2m-temperature mean absolute error are predominantly due to corresponding local changes in errors in the sea surface temperature and sea ice variables. There is no clear change in forecast skill of upper atmospheric circulation in our experiments. Our results illustrate that information on sea ice thickness is relevant for identifying model errors and for exploiting the long-term memory present in ice thickness

for seasonal forecasts of sea ice and near-surface temperatures. Constraining SIT in the initialisation alters biases arising due to both errors in the forcing and the sea-ice model. Though the SIT assimilation is not expected to solve these underlying problems per se, by moving the model state closer to reality, it helps us to better understand the errors in our system, as well as improving forecast skill scores in the meantime. As atmospheric forecast errors are dominated by biases, we are yet to demonstrate the benefit of interannual varying data on bias-corrected forecast scores. Robustness of impact on upper atmospheric variables and possible teleconnections need to be further assessed which would require a longer study period and larger sample size.

These findings demonstrate that making use of recently-available, spatially and temporally rich sea ice thickness observations from satellites for the ice growth season has the potential to significantly improve 1) the sea ice state in currently operational ocean–sea-ice reanalyses and, 2) the seasonal forecasts in operational forecasting systems. Our study also emphasizes the potential of future sea ice satellite missions for Earth System reanalysis and forecasts.

# Acknowledgements

We acknowledge the European Union Horizon 2020 SPICES (Space-borne observations for detecting and forecasting sea ice cover extremes) project (640161) for funding this work. MM's work was partially supported by the Austrian Science Fund project 33177. The production of the merged CryoSat2-SMOS sea

ice thickness data was funded by the ESA project SMOS and CryoSat-2 Sea Ice Data Product Processing and Dissemination Service, and data from 01/11/2011 to 31/12/2016 were obtained from Alfred Wegener Institute Helmholtz Centre for Polar and Marine Research (AWI).

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
