# Peer review of "Year-round Impact of Winter Sea Ice Thickness Observations on Seasonal Forecasts"

_The Cryosphere, 2020_

## Referee Comment (RC1) · Anonymous Referee #1 · 8 Jul 2020

Balan-Sarojini and co-authors present a study examining the impact of sea ice thickness (SIT) assimilation on seasonal forecasts of the northern hemisphere sea ice cover. In its approach and scope the study covers new ground; several of the key findings are substantive and represent a significant advance in our understanding of sea ice predictability and performance of seasonal-scale forecasts. The authors make good use of newly available, state-of-the-art ice thickness fields and strike a nice balance between more fundamental questions of prediction system performance, and applied questions related to improving forecast skills of Arctic sea ice models.

The paper is well structured and makes good use of figures to illustrate key points. The scientific approach is well described and appropriate for the problem at hand. The first half of the paper (up to and including Section 3.2, Fig. 7) is particularly compelling

and self-contained. The latter part of the manuscript, while interesting, is less compelling with some of the writing lacking clarity and the paper losing focus with respect to the goals laid out in the introduction and implicit in the title. If this part of the paper is retained in full, tightening the text and improving readability of sections 3.3-3.5 in particular would make the paper more accessible.

At the same time, a few aspects of the paper can be improved or require further thought, as outlined below.

First, given the central role the SMOS/Cryosat-2 data set plays in this study, one would like to see some discussion of how errors and uncertainties in the ice thickness data set may have impacted forecast skill and in particular some of the regional patterns observed in the thickness-assimilation runs. As shown in Ricker et al. (2017) uncertainties due to the fundamentally different retrieval approaches for SMOS and Cryosat-2, and to a lesser extent the optimal interpolation and data merging schemes, vary significantly by region. For example, over the Canadian Basin with mostly thick, multiyear ice the data product is dominated by bias/errors in Cryosat-2 data whereas in the Bering or Okhotsk Sea thinner ice weights errors towards those associated with SMOS thickness retrievals. It would be important to establish whether differences in thickness-field uncertainties have any impact on model performance and regional or temporal contrasts in forecast bias. This is also relevant for the integrated analyses of parameters such as the Integrated Ice Edge Error or ice volume at the pan-Arctic scale which may gloss over important regional contrasts in model performance.

Second, the paper lacks detail on the representation of ice thickness and key ice growth, melt and deformation processes in the LIM2 prognostic thermodynamic-dynamic sea ice model used in this study. It would be important to provide more detail, in particular as to whether any of the parameterizations that are part of the Fichefet & Morales Maqueda (1997) – FMM97 – model have been updated or changed. Of potential concern in FMM97 – based on description in their original paper – would be the limited representation of surface melt processes and their impact on ice albedo as

well as physically unrealistic representation of internal ice melt (with internal "storage" of solar heat up to a 50% threshold). These shortcomings, which may have been addressed in updates but if so the paper needs to make this explicit, do not necessarily limit applicability of the model in the context of seasonal ice forecasts. However, they are problematic in diagnosing some of the linkages between surface forcing, energy storage and the seasonal ice cycle explored in Section 3.3, since FMM97 in its original form may be ill suited to examine in particular the spring-summer-fall transitions in terms of the surface radiation balance or rates of ice thinning and decay.

Given these potential concerns, it would be instructive in Section 3.3 to examine the proportion of up/downwelling shortwave fluxes (or albedo, for that matter) to get a better perspective on the sensitivity of sea ice as represented in FMM97 to variations in downwelling shortwave energy. Such a detailed analysis may well be beyond the scope of the present paper. If so, this may be an argument to remove the latter parts of the paper as the basis for a separate, more detailed study. The first part of the paper (up to Section 3.3) is substantive enough and fully in line with the title of the paper.

Third, starting with the discussion of sea ice volume at the pan-Arctic or northern hemisphere scale the paper began to veer off-course a bit in terms of the goals laid out in the introduction. While total ice volume is a great integrator and a relevant variable in a global context, I was not able to tell whether the authors were assuming that it can also serve as an integrated measure of model performance in terms of ice concentration/extent and ice thickness. Given the regional contrasts in model performance apparent in the early figures of the paper the wholesale discussion of ice volume is somewhat problematic. For example, the interpretation of the seasonal ice volume cycle in terms of a single "freezing rate" (p. 17, top paragraph) is too simplistic since increases in ice volume during fall and winter occur through a combination of ice deformation and ice growth inside the ice pack as well as advance of the ice edge in marginal seas. Without an in-depth analysis some of the earlier figures and a solid understanding of how well the sea ice model is capturing the relevant processes, Figures such

as Fig. 8, don't add that much to the paper and could be relegated to supplemental materials or cut completely.

Finally, just a few minor points: - Comparing bias in ice thickness (Fig. 1) with bias in ice concentration (Fig. 3) it's striking that regions with near-zero bias in thickness (e.g. East Siberian Sea, Chukchi Sea in November) show up as having significant bias in ice concentration; moreover despite substantial contrasts in thickness biases between reference and ice thickness runs (Fig. 1c&d) the biases in ice concentration are near indistinguishable (Fig. 3 g&h). How can this be explained? - In regards to July ORA-SIT biases in ice concentration, it was striking to see much larger bias in the ORA-SIT than in the reference runs. Why would the simulations that performed (understandably) so much better in replicating ice thickness in March through assimilation of ice thickness data perform much worse in replicating ice concentration in July? Note that this finding also seems to contradict your statement in l. 185 that "The non-availability of the observations for the melt season in a way provides an opportunity to test the predictability of winter SIT from summer initial conditions." - You discuss your findings in terms of Arctic ice concentration and thickness but your figures include regions outside of the Arctic proper (such as the Okhotsk Sea). Please clarify whether both model output and assimilated data cover the entire northern hemisphere sea ice or a subset of that data. This is relevant in particular for figures like Fig. 5 which references "nh" in the figure label (for northern hemisphere?) but refers to Arctic sea ice area in the caption.

Minor comments & corrections

l. 2/3: change to "in its early stage"

l. 20 "near-surface temperature forecasts of early freezing season initialized in May": This phrase is confusing and not entirely clear, please revise to clarify what specifically is forecast with respect to "freezing season".

l. 25: change to "lasts into autumn"

l. 80: "it is relevant as cross-check variables evaluation" – not entirely clear what's referenced here – should it be "they are relevant because they allow for cross-checking between variables"? Please clarify.

l. 81: "SIT verification is also conducted as a sanity check of the nudging approach" – You lost me at "sanity check" – what exactly are you doing here? Please explain.

l. 91: change to "The Level-3"

l. 145: "LIM2 has a single sea ice category to represent sub-grid scale ice thickness distribution" – this needs further clarification. To calculate an effective conductive heat flux through the ice Fichefet and Morales Maqueda (1997) assumed a uniform thickness distribution bounded by zero and twice the average thickness. This parameterization was only applied in calculating heat fluxes through ice and lateral melt rate but did not enter into any of the ice dynamics components of the model. Given that ice thickness initialization is central to this manuscript, a clearer description of what exactly was implemented is needed.

l. 168: change to "differ in"

l. 233: "These results clearly show..." – Some clarification is needed here, since I interpret Fig. 4 as indicating that through May (but not the entire melt season), the effects of SIT assimilation are evident, beyond that the reference run performs better through the end of melt. In linking SIC increments to SIT assimilation please also consider the points raised in the general comments above.

l.238: "(units are..." – This should be part of the figure legend or caption, and not be buried in the main text.

l. 245: change to "melt season forecasts are considerably reduced"

l. 251: The top labels of the figure panels are cut off and it's not clear that they're actually needed ("bias for sia in area nh" – would need to be explained; also: is nh Northern Hemisphere? If so, what is the difference between this data for northern

hemisphere and the Arctic sea ice area as indicated in the figure caption?); the color scale needs better labeling.

l. 265: insert "are" in "that are to be expected"

l. 268: Fig. 6 - This figure should be cleaned up a bit as well; there's no need for two top labels (the upper one is more descriptive anyways, but even that's not needed given the explanation in the caption); the color bar needs proper units. Fig 7: Same comments apply – the 1e12 and 1e11 squeezed right next to the figure panel label and disjunct from the axis label (with units of square meters) are less than ideal and need to be cleaned up.

l. 287: Fig. 8: It's not clear to me how an axis label of $10^1 3$ mˆ3 translates into $10^{12}$ mˆ3 as the figure caption claims. Why not put an axis label in kmˆ3?

l. 361, Figure 11: same comments as for Fig 6 apply

l. 369: correct spelling of "Atlantic"

---

## Referee Comment (RC2) · Anonymous Referee #2 · 8 Jul 2020

########## # Summary

Balan-Sarojini et al. study the impact of Cryosat2/SMOS winter ice thickness (SIT) observation nudging on a lower-resolution version of the ECMWF ocean/sea-ice re-analysis (ORA) system and on associated coupled seasonal forecasts initialized from that reanalysis system. The SIT constraint suppresses an otherwise too strong annual SIT/SIV cycle in the ORA and provides overall thinner SIT conditions toward the end of the northern winter (except in the perennial ice regions north of Greenland and the CAA), which turn into decreased sea-ice extent in the ORA in summer (despite sea-ice concentration assimilation). The thinner/less extensive initial ice is benefitial for seasonal forecasts initialized before July, but forecasts initialised in late summer tend to be deteriorated. The authors show that this is linked to too-strong spring/summer melt in

the ORAs (when no SIT constraint is available), leading to low-biased ice and warm-biased sea-surface initial conditions in summer, in combination with a too-late/too-weak refreeze in the coupled forecast system. Balan-Sarojini et al. show evidence that the latter points can be explained at least partly with the surface radiation budget in the atmosphere-forced ORAs and in the coupled forecast model.

The study is scientifically sound, well-written, contains appropriate graphics and references, and provides interesting insights into the impact of ice thickness obserations on forecasts in the specific system used which might be helpful to understand other systems, too. I do have quite a number of remarks, most of which are however minor. The maybe most demanding recommendation is to compare against simple climatological benchmark forecasts where appropriate. In summary, I recommend publication of this work in The Cryosphere subject to minor(-to-major) revisions as detailed in the following.

########## # Specific comments

L12-13: "we find significant improvement of up to 28% in the September sea ice edge forecast started from April" - From the abstract it does not become clear that the paper is almost completely focussed on biases (and how these affected by constraining SIT) and not on interannual anomalies. In the summary section you state very clearly that this is the case (L441-442), but I think it should be mentioned in the abstract, too. Without that information, the sentence in L12-13 leaves one wondering how such a significant forecast improvement can be reconciled with the "May predictability barrier". In this context, see also my recommendation below to consider comparing with a climatological benchmark forecast where appropriate.

L57: Zampieri et al. 2018 - There's also a follow-on paper demonstrating reasonable skill of ECMWF S2S sea-ice forecasts in the Antarctic: Zampieri et al. 2019 "Predictability of Antarctic Sea Ice Edge on Subseasonal Time Scales".

Eq. 1: It probably doesn't make a big difference, but can you specify whether this

is "floe-thickness" or "effective thickess" (thickness when evenly distributed over grid cell)?

L162-164: "We have also tested the sensitivity to different nudging strengths by running variants of ORA-SIT with a relaxation time scale of 20, 30 and 60 days" - If you mention this, I would expect that you also say something about the impact of the relaxation timescale and why you chose 10 days.

L201-205: "slight underestimation over the central Arctic and overestimation over the Canadian Archipelago still remain in November. This is probably caused by the lack of SIT observations during the months preceeding November" - Given the relaxation timescale of 10 days, I assume that this difference goes back almost completely to the first half of November? That would confirm that you could omit the word "probably"; that's a rather obvious link.

L208-209: "The largest impact occurs in March, probably because at this month the SIT observations have been assimilated during the preceeding 5 months" - similar to what I say in the previous point, I assume that the SIT state responds according to the relaxation timescale. This implies that, on a monthly scale, the largest impact should occur in the month with the largest bias, no matter for how many months relaxation has been active before that month (as long as it's at least one month).

L210: "with a slight clockwise displacement" - you could mention that this is consistent with the mean climatological Arctic drift pattern (transpolar drift, Beaufort gyre) and thus likely a consequence of the mean advection.

L217-218: "In November [...] the SIT constraint has very little impact on SIC biases" - Could the reason be that (in addition to the fact that no SIT corrections are applied in the previous months) the thickness corrections made in Nov need more time to influence the sea-ice concentration, because that requires a "cross-impact" through other processes (dynamics and thermodynamics)?

L225: "large positive increments from May to October, indicative of strong underestimation of SIC in the ORAs" - To be precise, should "in the ORAs" rather be "in the (hypothetical) forced model without SIC assimilation"? After all, the SIC assimilation makes sure that the SIC underestimation doesn't get too strong.

L232-235: Isn't the even bigger difference in the SIC increments after May (even though these are for the worse) even more strongly showing the long-lasting impact of the SIT corrections on the SIC assimilation?

L243: "low bias" could be mistaken for "negative bias", maybe better say "weak bias" or "small bias" or similar

L250-262: To compute the IIEE, do you use the ensemble-median ice edge (50%-contour of sea-ice probability where SIC=15% is used to determine "presence" or "absence" of sea-ice in each ensemble member) or do you compute it for each member individually and average the IIEEs afterwards? That would make a difference, so this should be specified. Related, note that there's a probabilistic version of the IIEE ("Spatial Probability Score", Goessling and Jung 2018 "A probabilistic verification score for contours: Methodology and application to Arctic ice‐edge forecasts") that you could apply to your ensemble forecasts directly, which would have the advantage that changes in uncertainty/reliability would be captured, too.

Fig. 6 caption and throughout the paper: DelSole and Tippett (2016) just apply the sign test (a special case of the binomial test with p=0.5), only that they visualize how the outcome develops from forecast case to case like a random walk. I would recommend to refer to the test simply as the sign test (which in fact dates back to 1710!).

Sect. 3.2 and Fig. 7: 1) Can you please explain how the bias correction is performed? Is this simply done for each gridcell individually? Do you just subtract the mean concentration bias (difference as a function of time of the year and lead time, averaged over 2011-2016/17), possibly with a correction that makes sure concentration values remain bound between 0 and 1? Or is quantile normalization involved? 2) Related to

the bias correction, I would find it very useful if the forecast errors could be compared against a climatological benchmark forecast. The latter could be based simply on the same period (2011-2016/17), or on the preceding decade (to make it more "operational"). I would expect that the uncalibrated forecasts are worse than climatology for most lead times (except the first one or two months?), but the calibrated might beat the climatology for a few months? In the summary section you say very clearly that you are "yet to demonstrate the benefit of interannual varying data on bias-corrected forecast scores", but I think it would be rather easy and revealing to add a climatological reference (even if it reveals clear limitations of current sea-ice forecast skills).

Fig. 8, top: Can you provide an explanation why the SIV in the ORAs converge from May to September, so that the large SIT difference in spring is completely "forgotten", whereas the coupled forecasts maintain much of the initial offset? Is there some fundamental reason why the forced (vs. coupled) atmosphere would cause such a differenc, or can it be linked to the continued assimilation of SIC (or ocean variables)?

Eq. 2: The way the melt energy tendency is defined, is seems to be really just the derivative of (area-averaged) SIT (times a constant factor), right? Also, maybe it's better to use partial d's to make clear that these are not material (Lagrangian) derivatives. Related, you could also mention that changes in SIT through divergence as well as advection are included, implying that the "melt energy tendeny" can in principle also change through dynamics. I understand that, by averaging over a large area (almost the whole Arctic), most of any dynamical effects would be compensating each other, but being clear about this would be good.

L314-316 & Fig. 9: The plot caption reveals that for the forecasts you look only at the first-month MET, but you do not mention/explain/motivate this in the text. Further, do I understand correctly that, by considering just the first month of the respective forecasts instead of a "closed" seasonal cycle, the annual integral of MET is not expected to be zero (while it should be zero for the ORAs)? In fact it looks a bit like it's rather negative (average build-up of sea-ice volume), can you confirm this?

Fig. 10 and corresponding text: I am wondering to what extent turbulent fluxes (in particular sensible) could also play a role, for example, with stronger downward spring/summer sensible heat fluxes in the forced ORAs compared to the coupled forecasts (acknowledging that there might not be a corresponding observational dataset to compare against). Too high near-surface temperatures that could generate too strong downward sensible heat flux would be consistent with a positive downwelling longwave bias in ERA-I, even if clouds also seem to play a role there. If differences in turbulent fluxes are too small to be important, please mention that.

L351-352: "Significant cold biases are present in forecasts for most of the start months and lead months" - Is this also true over Arctic sea ice in winter? If so, how can it be reconciled with Batrak and Müller (2019) "On the warm bias in atmospheric reanalyses induced by the missing snow over Arctic sea-ice"? I thought that the surface coupling is similar in the system studied here?

Fig. 12: I was a few times slightly confused when looking at this figure, intuitively thinking that the lower panels show differences between FC-SIT and FC-REF that could be directly combined with the biases shown in the upper panels. But the lower panels show the differences in mean absolute error, which is alright but easily misleading. I suggest to use a different colourbar for the lower panels so that the different flavours of "temperature" (signed vs. unsigned) is more intuitively reflected.

########## # Technical corrections

L25: last -> lasts

L80: "as cross-check variables evaluation" - I recommend to reformulate.

L91: These -> This

L168: "differ on" -> "differ in" / "differ regarding"

L208: "gradients on" -> "gradient in the" or "gradients of the"

[Figure]

L212: "end of melt season" -> add "the"

L217: "reduced up to" -> "reduced by up to"

L227: "indicates" -> "indicate"

L228: "at marginal seas" -> "in the marginal seas"

L232: "on an average" -> "on average"

L232-233: "in ORA-SIT analysis" -> add "the"

L265: "that to be" -> add "are"

L288: "is smaller" -> "are smaller"

There are a few more such tiny things, please check carefully!

---

## Author Comment (AC1) · 28 Sep 2020

**Response to the review of tc-2020-73 titled "Year-round Impact of Winter Sea Ice Thickness Observations on Seasonal Forecasts"**

We are grateful to the two anonymous reviewers for the thorough review, insightful comments and generally positive response to our article titled "Year-round Impact of Winter Sea Ice Thickness Observations on Seasonal Forecasts". All of their remarks are addressed. Please find below our responses to both the reviewers' comments and suggestions (*in blue italics*).

**Response to Anonymous Referee #1**

*Balan-Sarojini and co-authors present a study examining the impact of sea ice thickness (SIT) assimilation on seasonal forecasts of the northern hemisphere sea ice cover. In its approach and scope the study covers new ground; several of the key findings are substantive and represent a significant advance in our understanding of sea ice predictability and performance of seasonal-scale forecasts. The authors make good use of newly available, state-of-the-art ice thickness fields and strike a nice balance between more fundamental questions of prediction system performance, and applied questions related to improving forecast skills of Arctic sea ice models. The paper is well structured and makes good use of figures to illustrate key points. The scientific approach is well described and appropriate for the problem at hand. The first half of the paper (up to and including Section 3.2, Fig. 7) is particularly compelling and self-contained. The latter part of the manuscript, while interesting, is less compelling with some of the writing lacking clarity and the paper losing focus with respect to the goals laid out in the introduction and implicit in the title. If this part of the paper is retained in full, tightening the text and improving readability of sections 3.3-3.5 in particular would make the paper more accessible.*

We thank the reviewer for the constructive comments on the article. Regarding sections 3.3-3.4 we propose to follow the reviewer's suggestion on improving the readability of these sections in the revised manuscript. We also justify the presence of these sections in the introduction by pointing out that this work takes a forecasting system end-to-end perspective, from observations, modelling to forecast products. Thus, in the revised version we clarify that the paper has three main foci sections, as the reviewer has noticed: i) assessing the seasonal forecasting system performance using new sea ice thickness (SIT) observational information (sections 3.1 and 3.2), ii) improving Arctic sea-ice forecast skill by understanding the errors in the coupled forecast model and the data assimilation system through targeted diagnostics (sections 3.3), and iii) quantifying the impact of sea-ice improvements on seasonal forecasts of atmospheric variables (section 3.4). We agree with the reviewer that sections 3.3 and 3.4 were not properly motivated in the introduction, and we intend to do so in the revised manuscript. Section 3.3 describes the inconsistency between the errors in the coupled forecast system and the analysis, a key result that points out that the dominant source of error lies in the atmospheric forcing rather than in the sea-ice model formulation or data assimilation, and indicates that improvement of atmospheric fluxes from atmospheric reanalyses is urgently needed to improve the Arctic sea-ice related forecasts. Section 3.4 explores the impact on forecast skill of atmospheric variables. Although seasonal predictions of sea ice can be an end by itself, a prime objective of ECMWF is the forecast of atmospheric variables. Therefore, a key part of the evaluation methodology for system developments includes the

verification of impact of atmospheric variables. In a first instance, this acts as a sanity check to make sure there are no obvious degradations, which adds robustness to the developments. In a second instance, it helps to quantify the impact on forecasts arising from small incremental improvements, which helps to put the SIT impact in the context of other model/data assimilation improvements. Hence, we would prefer to keep sections 3.3-3.4 in the same article for future reference. Please note that there is no section 3.5 in the article, and we believe it is a typo. Please also see our response to Reviewer 2's specific comment on L12-L13.

*At the same time, a few aspects of the paper can be improved or require further thought, as outlined below. First, given the central role the SMOS/Cryosat-2 data set plays in this study, one would like to see some discussion of how errors and uncertainties in the ice thickness data set may have impacted forecast skill and in particular some of the regional patterns observed in the thickness-assimilation runs. As shown in Ricker et al. (2017) uncertainties due to the fundamentally different retrieval approaches for SMOS and Cryosat2, and to a lesser extent the optimal interpolation and data merging schemes, vary significantly by region. For example, over the Canadian Basin with mostly thick, multiyear ice the data product is dominated by bias/errors in Cryosat-2 data whereas in the Bering or Okhotsk Sea thinner ice weights errors towards those associated with SMOS thickness retrievals. It would be important to establish whether differences in thickness-field uncertainties have any impact on model performance and regional or temporal contrasts in forecast bias. This is also relevant for the integrated analyses of parameters such as the Integrated Ice Edge Error or ice volume at the pan-Arctic scale which may gloss over important regional contrasts in model performance.*

We completely agree with the reviewer that the thickness uncertainties should be considered in sophisticated assimilation of SIT. However, in this work we have only taken the very first step in SIT assimilation by using a simple nudging method to constrain SIT from the merged Cryosat2-SMOS product without considering the observational uncertainties. In a merged product like CS2SMOS, it is difficult to represent the sensor-specific errors properly. As the reviewer commented, sensor-specific errors could affect regional SITs, i.e., over multiyear thick ice over the Canadian Basin, errors associated with Cryosat-2 retrievals dominate whereas in the Bering or Okhotsk Sea with mostly seasonal thin ice, errors associated with SMOS retrievals dominate. As reported in Ricker et al. 2017, the relative error is maximum in the thickness range of 0.5-1.0 m in the merged product where both CS2 thick ice and SMOS thin ice retrieval errors are maximum. We add this point in Section 2.1.2 and in the concluding section mentioning it as an area which needs to be explored in future studies of SIT assimilation. For instance, the uncertainty in the retrievals could be taken into account by perturbing the observations in the ensemble of data assimilations. We add a sentence on this aspect in the revised manuscript. Equally, the verification will benefit from having records of SIT with easy-to-use uncertainty estimates. The practice followed by HadISST2, which provides an ensemble of SST records to cover different sources of uncertainty in SST, is proving very convenient.

*Second, the paper lacks detail on the representation of ice thickness and key ice growth, melt and deformation processes in the LIM2 prognostic thermodynamic-dynamic sea ice model used in this study. It would be important to provide more detail, in particular as to whether any of the parameterizations that are part of the Fichefet & Morales Maqueda (1997) – FMM97 – model have been updated or*

*changed. Of potential concern in FMM97 – based on description in their original paper – would be the limited representation of surface melt processes and their impact on ice albedo as well as physically unrealistic representation of internal ice melt (with internal "storage" of solar heat up to a 50% threshold). These shortcomings, which may have been addressed in updates but if so the paper needs to make this explicit, do not necessarily limit applicability of the model in the context of seasonal ice forecasts. However, they are problematic in diagnosing some of the linkages between surface forcing, energy storage and the seasonal ice cycle explored in Section 3.3, since FMM97 in its original form may be ill suited to examine in particular the spring-summer-fall transitions in terms of the surface radiation balance or rates of ice thinning and decay. Given these potential concerns, it would be instructive in Section 3.3 to examine the proportion of up/downwelling shortwave fluxes (or albedo, for that matter) to get a better perspective on the sensitivity of sea ice as represented in FMM97 to variations in downwelling shortwave energy. Such a detailed analysis may well be beyond the scope of the present paper. If so, this may be an argument to remove the latter parts of the paper as the basis for a separate, more detailed study. The first part of the paper (up to Section 3.3) is substantive enough and fully in line with the title of the paper.*

We acknowledge that LIM2 is a simple sea ice model, which we have in our current operational system since 2017. The original version of FMM97 is used. As the reviewer points out, it has several limitations of surface melt processes such as for instance, representation of melt ponds which could affect the accurate representation of surface albedo over sea-ice. However, we note that only the ocean reanalysis ORAS5 actually makes use of the albedo computed by LIM2 (which is too high in summer), while the atmospheric reanalyses and the forecasting system use the same climatological albedo (based on SHEBA campaign observations; Beesley et al. 2000). This means that the differences found in Figure 10 cannot be attributed to different albedo biases in the atmospheric reanalyses and forecasts. We would also like to point out that a recent comparison study (Pohl et al. 2020) shows that, overall, the broadband albedo over Arctic sea-ice derived from MERIS observations is comparable to that in the ERA5 atmospheric reanalysis in terms of the seasonal cycle on the large spatial scales. We find that the forecast albedo over ice is comparable to that in ERA-5 and ERA-Interim atmospheric reanalyses. Moreover, it has been shown that the downwelling short wave radiation has a negative bias over the central Arctic in both the atmospheric reanalyses and the reforecasts (Hogan et al. 2017, Balan-Sarojini et al. 2019). We add the specific points in the model description part. We also emphasize that, although the manuscript does not cover sea-ice model improvements, recent developments in modelling sea-ice processes affecting the sea-ice                                                                                        melt/growth (https://forge.ipsl.jussieu.fr/nemo/chrome/site/doc/SI3/manual/pdf/SI3_manual.pdf)     are     being considered for inclusion in upcoming versions of the ECMWF forecasting systems.

*Third, starting with the discussion of sea ice volume at the pan-Arctic or northern hemisphere scale the paper began to veer off-course a bit in terms of the goals laid out in the introduction. While total ice volume is a great integrator and a relevant variable in a global context, I was not able to tell whether the authors were assuming that it can also serve as an integrated measure of model performance in terms of ice concentration/extent and ice thickness. Given the regional contrasts in model performance apparent in the early figures of the paper the wholesale discussion of ice volume is somewhat problematic. For*

*example, the interpretation of the seasonal ice volume cycle in terms of a single "freezing rate" (p. 17, top paragraph) is too simplistic since increases in ice volume during fall and winter occur through a combination of ice deformation and ice growth inside the ice pack as well as advance of the ice edge in marginal seas. Without an in-depth analysis some of the earlier figures and a solid understanding of how well the sea ice model is capturing the relevant processes, Figures such as Fig. 8, don't add that much to the paper and could be relegated to supplemental materials or cut completely.*

We agree to the reviewer that an integrated quantity like Arctic sea ice volume is a result of many dynamic and thermodynamic sea-ice processes and lacks regional details. However, integrated SIV is a key indicator for understanding of the Arctic energy cycle, an important process that needs to be represented in reanalyses and seasonal forecast. It is useful to compare the contrasting SIV seasonal cycles in coupled and uncoupled mode, and with/without SIT observational constraint in the initialization, since this helps to identify the origin of errors in the systems in the specific operational set up. As noted in L304-305, SIC increments in the ORAs do affect analyzed SIV. We add a few sentences to discuss the benefits and caveats of using pan-Arctic sea-ice volume as a diagnostic for model performance in the revised manuscript.

*Finally, just a few minor points: - Comparing bias in ice thickness (Fig. 1) with bias in ice concentration (Fig. 3) it's striking that regions with near-zero bias in thickness (e.g. East Siberian Sea, Chukchi Sea in November) show up as having significant bias in ice concentration; moreover despite substantial contrasts in thickness biases between reference and ice thickness runs (Fig. 1c&d) the biases in ice concentration are near indistinguishable (Fig. 3 g&h). How can this be explained?*

We thank the reviewer for raising this important question, which we failed to comment in the original version. The sea-ice thickness bias in ORA-SIT in these areas is very small, about 0.1 to 0.05 m (below the contour interval). The presence of concentration bias (Figure 3, similar pattern in ORA-SIT and ORA-REF) in regions with negligible thickness bias in ORA-SIT is suggestive of fast growth processes in the forward model, faster than the timescales intrinsic to the assimilation of sea-ice concentration. We add this explanation in the revised manuscript.

*- In regards to July ORASIT biases in ice concentration, it was striking to see much larger bias in the ORA-SIT than in the reference runs. Why would the simulations that performed (understandably) so much better in replicating ice thickness in March through assimilation of ice thickness data perform much worse in replicating ice concentration in July?*

A very pertinent question, we asked ourselves as well. As noted in L182-L185 of original text, sea-ice thickness observations are only available and assimilated for November-March. The ORA-SIT sea ice thickness from April-October are not constrained by observations. The fact that ORA-SIT has larger errors than ORA-REF in SIC for July is attributed to the overestimation of the melt in the forced model, as discussed in the last two paragraphs of section 3.1. The negative summer SIC bias gets worse in ORA-SIT than that in ORA-REF due to the fact that the ORA-SIT starts from a thinner ice state compared to ORA-REF without CS2SMOS thinning. Please also read our response to L208-209 in the Reviewer2's response part. Indeed, the assimilation of sea-ice concentration is trying hard to compensate for this

excess of sea-ice melt (Figure 4). The reasons for this excess sea-ice melt during this season is further investigated in section 3.3 and attributed to errors in forcing fluxes. This is one of the main outcomes of this work. We revise the manuscript to bring out the answer to the reviewer's question more clearly by adding the above discussion in Section 4.

*Note that this finding also seems to contradict your statement in l. 185 that "The non-availability of the observations for the melt season in a way provides an opportunity to test the predictability of winter SIT from summer initial conditions."*

Thanks for this point. We agree that the sentence is confusing and we remove it in the revised manuscript.

*You discuss your findings in terms of Arctic ice concentration and thickness but your figures include regions outside of the Arctic proper (such as the Okhotsk Sea). Please clarify whether both model output and assimilated data cover the entire northern hemisphere sea ice or a subset of that data. This is relevant in particular for figures like Fig. 5 which references "nh" in the figure label (for northern hemisphere?) but refers to Arctic sea ice area in the caption.*

Thanks for pointing out the confusion on the definition of Arctic domain in the article. We would like to clarify that we have used a global model, so its output includes the entire northern hemisphere. And we exploit the full spatial coverage of CS2SMOS data set, which covers all the regions where sea ice has ever been observed in recent decades, so it can safely be treated as representing the entire northern hemisphere. This information is clearly stated in S2.1.2 and S2.2.1 in the revised manuscript.

We also add a definition of pan-Arctic (as the sea ice area included is of the whole of NH which is essentially the Arctic sea ice and the Baltic sea ice) in the beginning of the Results section and use the term 'pan-Arctic' wherever appropriate except for Figures where we have explicitly mentioned the Arctic domain area in the caption. In this work, we are interested in the large-scale impact and not in grid-point scale impact.

**Response to Minor comments & corrections**

*l. 2/3: change to "in its early stage"*

thanks, it is done.

*l. 20 "near-surface temperature forecasts of early freezing season initialized in May": This phrase is confusing and not entirely clear, please revise to clarify what specifically is forecast with respect to "freezing season".*

thanks, it is rephrased as "near-surface temperature forecasts of early freezing season (Sept-Oct-Nov) initialized in May".

*l. 25: change to "lasts into autumn"*

thanks, it is done.

*l. 80: "it is relevant as cross-check variables evaluation" – not entirely clear what's referenced here – should it be "they are relevant because they allow for cross-checking between variables"? Please clarify.*

thanks for pointing it, the reviewer is right. It is rephrased as "Although the datasets used for verification are not strictly independent, evaluation using those datasets is relevant as it allows cross-checking between variables, for instance between SIC and SIT assimilation.".

*l. 81: "SIT verification is also conducted as a sanity check of the nudging approach" – You lost me at "sanity check" – what exactly are you doing here? Please explain.*

By 'sanity check' we meant that SIT verification using CS2SMOS dataset (Figure 1) is a basic test to check whether the nudging works in the first place. L81 of the original text is rephrased as "SIT verification using CS2SMOS dataset is also conducted as a sanity check of the nudging approach: the approach works, the differences with respect to CS2SMOS should be smaller in ORA-SIT than in ORA-REF.".

*l. 91: change to "The Level-3"*

thanks, it is done.

*l. 145: "LIM2 has a single sea ice category to represent sub-grid scale ice thickness distribution" – this needs further clarification. To calculate an effective conductive heat flux through the ice Fichefet and Morales Maqueda (1997) assumed a uniform thickness distribution bounded by zero and twice the average thickness. This parameterization was only applied in calculating heat fluxes through ice and lateral melt rate but did not enter into any of the ice dynamics components of the model. Given that ice thickness initialization is central to this manuscript, a clearer description of what exactly was implemented is needed.*

The reviewer is right. As we responded earlier in the main comments section, the original version of Fichefet and Morales Maqueda (1997) LIM2 version is implemented in our operational system. We explicitly mention it in the revised version.

*l. 168: change to "differ in"*

thanks, it is corrected.

*l. 233: "These results clearly show. . ." – Some clarification is needed here, since I interpret Fig. 4 as indicating that through May (but not the entire melt season), the effects of SIT assimilation are evident, beyond that the reference run performs better through the end of melt. In linking SIC increments to SIT assimilation please also consider the points raised in the general comments above.*

Thanks for pointing it out. We agree with the reviewer that a positive impact on SIT is seen till May and a negative impact is seen till September. As we have already described the nature of impact in the preceding paragraph, this is a general summary sentence at the end of the subsection. To avoid confusion we rephrase the sentence as "These results clearly show the long-lasting effect of the SIT information: the SIT constraint was only applied during the growth season from November to March

(grey shading), but its impact, whether positive or negative, is evident in sea ice concentration throughout the melting season even in the presence of SIC assimilation".

*l.238: "(units are. . ." – This should be part of the figure legend or caption, and not be buried in the main text.*

The caption has it already. We remove it here, thanks.

*l. 245: change to "melt season forecasts are considerably reduced"*

thanks, it is done.

*l. 251: The top labels of the figure panels are cut off and it's not clear that they're actually needed ("bias for sia in area nh" – would need to be explained; also: is nh Northern Hemisphere? If so, what is the difference between this data for northern hemisphere and the Arctic sea ice area as indicated in the figure caption?); the color scale needs better labeling.*

Thanks, we confirm that the Arctic domain we have considered everywhere, unless it is specifically mentioned in the Figure captions, are pan Arctic which is defined in the revised Results section too. We remove the confusing term in the figure panel in the revised paper.

*l. 265: insert "are" in "that are to be expected"*

thanks, it is done.

*l. 268: Fig. 6 - This figure should be cleaned up a bit as well; there's no need for two top labels (the upper one is more descriptive anyways, but even that's not needed given the explanation in the caption); the color bar needs proper units. Fig 7: Same comments apply – the 1e12 and 1e11 squeezed right next to the figure panel label and disjunct from the axis label (with units of square meters) are less than ideal and need to be cleaned up.*

thanks for the suggestion, it is done.

*l. 287: Fig. 8: It's not clear to me how an axis label of 10ˆ1 3 mˆ3 translates into 10ˆ12 mˆ3 as the figure caption claims. Why not put an axis label in kmˆ3?*

thanks, it is done.

*l. 361, Figure 11: same comments as for Fig 6 apply*

thanks, it is done.

*l. 369: correct spelling of "Atlantic"*

the typo is corrected.

**Response to Anonymous Referee #2**

*Balan-Sarojini et al. study the impact of Cryosat2/SMOS winter ice thickness (SIT) observation nudging on a lower-resolution version of the ECMWF ocean/sea-ice reanalysis (ORA) system and on associated coupled seasonal forecasts initialized from that reanalysis system. The SIT constraint suppresses an otherwise too strong annual SIT/SIV cycle in the ORA and provides overall thinner SIT conditions toward the end of the northern winter (except in the perennial ice regions north of Greenland and the CAA), which turn into decreased sea-ice extent in the ORA in summer (despite sea-ice concentration assimilation). The thinner/less extensive initial ice is benefial for seasonal forecasts initialized before July, but forecasts initialised in late summer tend to be deteriorated. The authors show that this is linked to too-strong spring/summer melt in the ORAs (when no SIT constraint is available), leading to low-biased ice and warm biased sea-surface initial conditions in summer, in combination with a too-late/too-weak refreeze in the coupled forecast system. Balan-Sarojini et al. show evidence that the latter points can be explained at least partly with the surface radiation budget in the atmosphere-forced ORAs and in the coupled forecast model. The study is scientifically sound, well-written, contains appropriate graphics and references, and provides interesting insights into the impact of ice thickness observations on forecasts in the specific system used which might be helpful to understand other systems, too. I do have quite a number of remarks, most of which are however minor. The maybe most demanding recommendation is to compare against simple climatological benchmark forecasts where appropriate. In summary, I recommend publication of this work in The Cryosphere subject to minor(-to-major) revisions as detailed in the following.*

We thank the reviewer for the positive remarks on our article. The main suggestion to verify the reforecasts against a climatological benchmark forecast is appreciated. Comparison against a climatological benchmark forecast is a very useful background information for evaluating the predictive skill of multi-model-ensemble forecasting systems (for example as in Zampieri et al. 2018), and we add it in the revised manuscript, even if benchmarking dynamical seasonal forecast against climatology is not the main objective of the paper.

**Response to Specific comments**

*L12-13: "we find significant improvement of up to 28% in the September sea ice edge forecast started from April" - From the abstract it does not become clear that the paper is almost completely focussed on biases (and how these affected by constraining SIT) and not on interannual anomalies. In the summary section you state very clearly that this is the case (L441-442), but I think it should be mentioned in the abstract, too. Without that information, the sentence in L12-13 leaves one wondering how such a significant forecast improvement can be reconciled with the "May predictability barrier". In this context, see also my recommendation below to consider comparing with a climatological benchmark forecast where appropriate.*

Thanks for pointing it out. We mention in the revised abstract that change in biases is the main impact.

*L57: Zampieri et al. 2018 - There's also a follow-on paper demonstrating reasonable skill of ECMWF S2S sea-ice forecasts in the Antarctic: Zampieri et al. 2019 "Predictability of Antarctic Sea Ice Edge on Subseasonal Time Scales".*

We add here the reference of the suggested paper on the Antarctic sea ice skill too.

*Eq. 1: It probably doesn't make a big difference, but can you specify whether this is "floe-thickness" or "effective thickness" (thickness when evenly distributed over grid cell)?*

Thanks, we mention that it is the "floe thickness".

*L162-164: "We have also tested the sensitivity to different nudging strengths by running variants of ORA-SIT with a relaxation time scale of 20, 30 and 60 days" - If you mention this, I would expect that you also say something about the impact of the relaxation timescale and why you chose 10 days.*

As the relaxation time scale increases from 10 days to 60 days, lesser constraint on SIT is found. We chose the time scale of 10 days for 2 reasons: 1) it fits to the length of the assimilation window, and 2) we first wanted to look at the forecast impact of the initial conditions with the maximum observational constraint. We add a line on this point.

*L201-205: "slight underestimation over the central Arctic and overestimation over the Canadian Archipelago still remain in November. This is probably caused by the lack of SIT observations during the months preceeding November" - Given the relaxation timescale of 10 days, I assume that this difference goes back almost completely to the first half of November? That would confirm that you could omit the word "probably"; that's a rather obvious link.*

We agree with the reviewer. We could remove "probably". Please also see our response to the next remark.

*L208-209: "The largest impact occurs in March, probably because at this month the SIT observations have been assimilated during the preceeding 5 months" - similar to what I say in the previous point, I assume that the SIT state responds according to the relaxation timescale. This implies that, on a monthly scale, the largest impact should occur in the month with the largest bias, no matter for how many months relaxation has been active before that month (as long as it's at least one month).*

Thanks for raising this point. We agree that the relaxation timescale sets the degree of observational constraint as expected and that the largest impact occurs in the month with the largest bias. The reviewer could be right on the last point. But we can only confirm that statement after conducting assimilation experiments with each month observationally constraint as if the observations were only available for that particular month. Indeed, this is something we want to experiment in the future. So we would prefer to keep the word "probably".

*L210: "with a slight clockwise displacement" - you could mention that this is consistent with the mean climatological Arctic drift pattern (transpolar drift, Beaufort gyre) and thus likely a consequence of the mean advection.*

Thanks, we add this point.

*L217-218: "In November [...] the SIT constraint has very little impact on SIC biases" - Could the reason be that (in addition to the fact that no SIT corrections are applied in the previous months) the thickness corrections made in Nov need more time to influence the sea-ice concentration, because that requires a "cross-impact" through other processes (dynamics and thermodynamics)?*

As explained in our response to Reviewer 1's related comment starting with "Finally, a few minor comments", we now explain the seasonal cycle of the differences in SIC bias better in the revised manuscript. Firstly, there is no SIT nudging happening in the preceding months. Secondly, the negligible changes in SIC bias between ORA-REF and ORA-SIT is suggestive of fast growth processes in the forward model which is faster than the timescales intrinsic to the SIC assimilation. We provide this explanation in the revised manuscript.

*L225: "large positive increments from May to October, indicative of strong underestimation of SIC in the ORAs" - To be precise, should "in the ORAs" rather be "in the (hypothetical) forced model without SIC assimilation"? After all, the SIC assimilation makes sure that the SIC underestimation doesn't get too strong.*

Thanks for the suggestion. Indeed, the assimilation of SIC reduces the errors in concentration, that would be otherwise larger. We modify the sentence as " …indicative of the strong underestimation of SIC in the forward model…".

*L232-235: Isn't the even bigger difference in the SIC increments after May (even though these are for the worse) even more strongly showing the long-lasting impact of the SIT corrections on the SIC assimilation?*

The reviewer is right.

*L243: "low bias" could be mistaken for "negative bias", maybe better say "weak bias" or "small bias" or similar*

That is true, thanks, it is changed to "small bias".

*L250-262: To compute the IIEE, do you use the ensemble-median ice edge (50%- contour of sea-ice probability where SIC=15% is used to determine "presence" or "absence" of sea-ice in each ensemble member) or do you compute it for each member individually and average the IIEEs afterwards? That would make a difference, so this should be specified. Related, note that there's a probabilistic version of the IIEE ("Spatial Probability Score", Goessling and Jung 2018 "A probabilistic verification score for contours: Methodology and application to Arctic iceâAˇ Redge forecasts") that you ˇ could apply to your ensemble forecasts directly, which would have the advantage that changes in uncertainty/reliability would be captured, too.*

For simplicity, we compute the IIEE for the ice edge of the ensemble mean. Thanks for the suggestion on the SPS metric. We appreciate that different possibilities of computing IIEE give different results, and the SPS again can give a different result. However, in light of the large differences between the forecasts in

the present study, the differences are probably small. We test these other approaches and document in the revised paper whether they would lead to noticeable differences in the figures.

*Fig. 6 caption and throughout the paper: DelSole and Tippett (2016) just apply the sign test (a special case of the binomial test with p=0.5), only that they visualize how the outcome develops from forecast case to case like a random walk. I would recommend to refer to the test simply as the sign test (which in fact dates back to 1710!).*

We thank the reviewer for pointing us to the historical roots of this test. We would like to keep citing DelSole and Tippet (2016) as the most recent and most relevant piece of work in applying and refining this long-known test for the field of climate and weather forecasts. We follow the reviewer's suggestion to refer to it as the sign test (also in Figures 7, 11 and 12 captions) and modify the text as "...sign test as recommended by DelSole and Tippet (2016)".

*Sect. 3.2 and Fig. 7: 1) Can you please explain how the bias correction is performed? Is this simply done for each gridcell individually? Do you just subtract the mean concentration bias (difference as a function of time of the year and lead time, averaged over 2011-2016/17), possibly with a correction that makes sure concentration values remain bound between 0 and 1? Or is quantile normalization involved? 2) Related to the bias correction, I would find it very useful if the forecast errors could be compared against a climatological benchmark forecast. The latter could be based simply on the same period (2011-2016/17), or on the preceding decade (to make it more "operational"). I would expect that the uncalibrated forecasts are worse than climatology for most lead times (except the first one or two months?), but the calibrated might beat the climatology for a few months? In the summary section you say very clearly that you are "yet to demonstrate the benefit of interannual varying data on bias-corrected forecast scores", but I think it would be rather easy and revealing to add a climatological reference (even if it reveals clear limitations of current sea-ice forecast skills).*

1) We perform a simple bias correction like so: for each combination of grid cell, start date and forecast lead time, we compute the mean forecast error over all forecasts, which is then subtracted from the "raw" forecast value in order to obtain the bias-corrected forecast value. We do not clip the bias-corrected forecast values to make sure they are between 0 and 1. Although this should be done when issuing forecasts, Johnson et al. (2018) have shown that it makes negligible difference for forecast skill assessment.

2) We appreciate the comparison to a climatological reference forecast is an interesting point, and we include the climatological reference in Figure 7. However, we do not plan to dwell on this point, since this is not the main point of the work, and including further discussion on the performance of climatology would distract the readers from the main point: to determine whether initialization with CS2SMOS improves or deteriorates the forecast.

*Fig. 8, top: Can you provide an explanation why the SIV in the ORAs converge from May to September, so that the large SIT difference in spring is completely "forgotten", whereas the coupled forecasts maintain much of the initial offset? Is there some fundamental reason why the forced (vs. coupled) atmosphere*

*would cause such a difference, or can it be linked to the continued assimilation of SIC (or ocean variables)?*

As the reviewer suspects, the explanation for the different behavior of the ORAs and the forecasts is that the ORAs are constrained by the same SIC (but no SIT) information in summer, which leads to the convergence of the sea ice state in the ORAs during that time of the year. This effect can also be appreciated from Figure 4, which shows that the SIC assimilation increments of ORA-SIT in summer are much more positive than those of ORA-REF, suggesting that SIC assimilation needs to work harder to keep ORA-SIT on track (compared to ORA-REF), but overall acts to bring the ORAs closer together in the absence of ORA-SIT information.

In contrast, the coupled forecasts do not have a similar constraint and thus tend to keep the offset in the initial conditions throughout the forecast. However, Figure 8a shows that FC-SIT and FC-REF tend to converge as well, although much more slowly than the ORAs. This is mentioned in the revised text.

*Eq. 2: The way the melt energy tendency is defined, is seems to be really just the derivative of (area-averaged) SIT (times a constant factor), right? Also, maybe it's better to use partial d's to make clear that these are not material (Lagrangian) derivatives. Related, you could also mention that changes in SIT through divergence as well as advection are included, implying that the "melt energy tendency" can in principle also change through dynamics. I understand that, by averaging over a large area (almost the whole Arctic), most of any dynamical effects would be compensating each other, but being clear about this would be good.*

Yes, MET is simply proportional to temporal changes in effective sea ice thickness. As the reviewer points out, MET can also change dynamically through lateral ice transports, but here we average over all ocean north of 70N, which should be a sufficiently large area to average out this effect and should mainly leave thermodynamic effects as the drivers of MET. We clarify this point and also use the more appropriate partial d's in the revised manuscript as the reviewer suggested.

*L314-316 & Fig. 9: The plot caption reveals that for the forecasts you look only at the first-month MET, but you do not mention/explain/motivate this in the text. Further, do I understand correctly that, by considering just the first month of the respective forecasts instead of a "closed" seasonal cycle, the annual integral of MET is not expected to be zero (while it should be zero for the ORAs)? In fact it looks a bit like it's rather negative (average build-up of sea-ice volume), can you confirm this?*

In Figure 9 we want to isolate the changes in MET when switching from forced (=analysis) to coupled (= forecast) mode. To avoid seeing mainly the effect of feedbacks arising from the model sea ice state drifting away from the analyzed state (most notably the ice-albedo feedback), we decided to compile the annual cycle of forecasted MET from lead-month 1 data. This motivation is clarified in the revised manuscript.

The reviewer is also right that the average of the MET annual cycle in the ORAs is close to zero (in fact about ~+0.3 Wm-2, in agreement with the long-term sea ice melt), while it is ~-4.8Wm-2 in FC-REF. The

negative value suggests that lead-month 1 forecasts on average produce too much ice in winter and melt too little ice in summer. This point is noted in the revised manuscript.

*Fig. 10 and corresponding text: I am wondering to what extent turbulent fluxes (in particular sensible) could also play a role, for example, with stronger downward spring/summer sensible heat fluxes in the forced ORAs compared to the coupled forecasts (acknowledging that there might not be a corresponding observational dataset to compare against). Too high near-surface temperatures that could generate too strong downward sensible heat flux would be consistent with a positive downwelling longwave bias in ERA-I, even if clouds also seem to play a role there. If differences in turbulent fluxes are too small to be important, please mention that.*

Sea ice and near-surface air temperature are close to 0°C during the melting season in both our reanalyses and forecasts. Because of this weak vertical temperature gradient sensible heat fluxes will be generally small over sea ice in summer. Nevertheless, to be sure, we checked mean difference in sensible heat fluxes in ERA-Interim and the forecasts and differences over sea ice were uniformly small (<2Wm-2 for May and July averages), confirming that differences in this field cannot explain the found differences in MET. A short note on this is added to the revised manuscript.

*L351-352: "Significant cold biases are present in forecasts for most of the start months and lead months" - Is this also true over Arctic sea ice in winter? If so, how can it be reconciled with Batrak and Müller (2019) "On the warm bias in atmospheric reanalyses induced by the missing snow over Arctic sea-ice"? I thought that the surface coupling is similar in the system studied here?*

Yes, cold biases in near-surface temperature (T2m) are present in the forecasts over Arctic sea ice in winter when considering ERA5 reanalysis as the truth. Batrak and Mueller's findings on warm biases in sea-ice temperature in a group of atmospheric reanalyses (including ERA5) without realistic representations of snow over sea ice, and sea ice thickness, is based on verification against observations, and reanalysis products. The reviewer is right that using atmospheric or ocean reanalysis for verification of Arctic surface temperature can be misleading, since there is large uncertainty in them, as Batrak and Mueller 2019 show in their Figure 3. Verifying against observations is not easy, since due to the scarcity of observational campaigns over sea ice, the verification will have large representativeness error, and definitely not suitable for seasonal forecasts. So, while it is clear that assimilation of SIT has a sizeable and significant impact on T2m forecasts via SIC forecasts, we do not have enough information to assess if this contributes to the reduction of the mean error in T2m. We modify the manuscript along these lines.

*Fig. 12: I was a few times slightly confused when looking at this figure, intuitively thinking that the lower panels show differences between FC-SIT and FC-REF that could be directly combined with the biases shown in the upper panels. But the lower panels show the differences in mean absolute error, which is alright but easily misleading. I suggest to use a different colourbar for the lower panels so that the different flavours of "temperature" (signed vs. unsigned) is more intuitively reflected.*

We change to a different color bar in the lower panels where the differences in MAE are shown.

**Response to Minor comments & corrections**

*L25: last -> lasts*

thanks, it is done.

*L80: "as cross-check variables evaluation" - I recommend to reformulate.*

Thanks. As both the reviewers pointed out, it is rephrased as "Although the datasets used for verification are not strictly independent, evaluation using those datasets is relevant as it allows cross-checking between variables, for instance between SIC and SIT assimilation. "

*L91: These -> This*

thanks, the grammar is corrected.

*L168: "differ on" -> "differ in" / "differ regarding"*

thanks, it is replaced with "differ in".

*L208: "gradients on" -> "gradient in the" or "gradients of the"*

thanks, it is replaced with "gradient in the".

*L212: "end of melt season" -> add "the"*

thanks, it is done.

*L217: "reduced up to" -> "reduced by up to"*

thanks, it is done.

*L227: "indicates" -> "indicate"*

thanks, it is done.

*L228: "at marginal seas" -> "in the marginal seas"*

thanks, it is done.

*L232: "on an average" -> "on average"*

sorry, "on an average" is correct.

*L232-233: "in ORA-SIT analysis" -> add "the"*

thanks, it is done.

*L265: "that to be" -> add "are"*

thanks, it is done.

*L288: "is smaller" -> "are smaller"*

thanks, it is done.

*There are a few more such tiny things, please check carefully!*

thanks, we check the revised manuscript and correct where necessary.

**References**

Pohl, C., Istomina, L., Tietsche, S., Jaekel, E., Stapf, J., Spreen, G., and Heygster, G.: Broad Band Albedo of Arctic Sea Ice from MERIS Optical Data. The Cryosphere, 14, 165-182, 2020.

Hogan, R., Ahlgrimm, M., Balsamo, G., Beljaars, A., Berrisford, P., Bozzo, A., pe, F. D. G., Forbes, R., Haiden, T., Lang, S., Mayer, M., Polichtchouk, I., Sandu, I., Vitart, F., and Wedi, N.: Radiation in numerical weather prediction, ECMWF Technical Memorandum, https://doi.org/10.21957/2bd5dkj8x, https://www.ecmwf.int/node/17771, 2017.

Balan-Sarojini, B., Tietsche, S., Mayer, M., 465 Balmaseda, M., and Zuo, H.: Towards improved sea ice initialization and forecasting with the IFS, https://doi.org/10.21957/mt6m6rpwt, https://www.ecmwf.int/node/18918, 2019.

Zampieri, L., Goessling, H. F., and Jung, T.: Bright prospects for Arctic sea ice prediction on subseasonal time scales, Geophysical Research Letters, 45, 9731–9738, 2018.

---

## Editor Decision (ED1)

########## # Summary

Balan-Sarojini et al. study the impact of Cryosat2/SMOS winter ice thickness (SIT) observation nudging on a lower-resolution version of the ECMWF ocean/sea-ice re-analysis (ORA) system and on associated coupled seasonal forecasts initialized from that reanalysis system. The SIT constraint suppresses an otherwise too strong annual SIT/SIV cycle in the ORA and provides overall thinner SIT conditions toward the end of the northern winter (except in the perennial ice regions north of Greenland and the CAA), which turn into decreased sea-ice extent in the ORA in summer (despite sea-ice concentration assimilation). The thinner/less extensive initial ice is benefitial for seasonal forecasts initialized before July, but forecasts initialised in late summer tend to be deteriorated. The authors show that this is linked to too-strong spring/summer melt in

the ORAs (when no SIT constraint is available), leading to low-biased ice and warm-biased sea-surface initial conditions in summer, in combination with a too-late/too-weak refreeze in the coupled forecast system. Balan-Sarojini et al. show evidence that the latter points can be explained at least partly with the surface radiation budget in the atmosphere-forced ORAs and in the coupled forecast model.

The study is scientifically sound, well-written, contains appropriate graphics and references, and provides interesting insights into the impact of ice thickness obserations on forecasts in the specific system used which might be helpful to understand other systems, too. I do have quite a number of remarks, most of which are however minor. The maybe most demanding recommendation is to compare against simple climatological benchmark forecasts where appropriate. In summary, I recommend publication of this work in The Cryosphere subject to minor(-to-major) revisions as detailed in the following.

########## # Specific comments

L12-13: "we find significant improvement of up to 28% in the September sea ice edge forecast started from April" - From the abstract it does not become clear that the paper is almost completely focussed on biases (and how these affected by constraining SIT) and not on interannual anomalies. In the summary section you state very clearly that this is the case (L441-442), but I think it should be mentioned in the abstract, too. Without that information, the sentence in L12-13 leaves one wondering how such a significant forecast improvement can be reconciled with the "May predictability barrier". In this context, see also my recommendation below to consider comparing with a climatological benchmark forecast where appropriate.

L57: Zampieri et al. 2018 - There's also a follow-on paper demonstrating reasonable skill of ECMWF S2S sea-ice forecasts in the Antarctic: Zampieri et al. 2019 "Predictability of Antarctic Sea Ice Edge on Subseasonal Time Scales".

Eq. 1: It probably doesn't make a big difference, but can you specify whether this

is "floe-thickness" or "effective thickess" (thickness when evenly distributed over grid cell)?

L162-164: "We have also tested the sensitivity to different nudging strengths by running variants of ORA-SIT with a relaxation time scale of 20, 30 and 60 days" - If you mention this, I would expect that you also say something about the impact of the relexation timescale and why you chose 10 days.

L201-205: "slight underestimation over the central Arctic and overestimation over the Canadian Archipelago still remain in November. This is probably caused by the lack of SIT observations during the months preceeding November" - Given the relaxation timescale of 10 days, I assume that this difference goes back almost completely to the first half of November? That would confirm that you could omit the word "probably"; that's a rather obvious link.

L208-209: "The largest impact occurs in March, probably because at this month the SIT observations have been assimilated during the preceeding 5 months" - similar to what I say in the previous point, I assume that the SIT state responds according to the relaxation timescale. This implies that, on a monthly scale, the largest impact should occur in the month with the largest bias, no matter for how many months relaxation has been active before that month (as long as it's at least one month).

L210: "with a slight clockwise displacement" - you could mention that this is consistent with the mean climatological Arctic drift pattern (transpolar drift, Beaufort gyre) and thus likely a consequence of the mean advection.

L217-218: "In November [...] the SIT constraint has very little impact on SIC biases" - Could the reason be that (in addition to the fact that no SIT corrections are applied in the previous months) the thickness corrections made in Nov need more time to influence the sea-ice concentration, because that requires a "cross-impact" through other processes (dynamics and thermodynamics)?

L225: "large positive increments from May to October, indicative of strong underestimation of SIC in the ORAs" - To be precise, should "in the ORAs" rather be "in the (hypothetical) forced model without SIC assimilation"? After all, the SIC assimilation makes sure that the SIC underestimation doesn't get too strong.

L232-235: Isn't the even bigger difference in the SIC increments after May (even though these are for the worse) even more strongly showing the long-lasting impact of the SIT corrections on the SIC assimilation?

L243: "low bias" could be mistaken for "negative bias", maybe better say "weak bias" or "small bias" or similar

L250-262: To compute the IIEE, do you use the ensemble-median ice edge (50%-contour of sea-ice probability where SIC=15% is used to determine "presence" or "absence" of sea-ice in each ensemble member) or do you compute it for each member individually and average the IIEEs afterwards? That would make a difference, so this should be specified. Related, note that there's a probabilistic version of the IIEE ("Spatial Probability Score", Goessling and Jung 2018 "A probabilistic verification score for contours: Methodology and application to Arctic ice‐edge forecasts") that you could apply to your ensemble forecasts directly, which would have the advantage that changes in uncertainty/reliability would be captured, too.

Fig. 6 caption and throughout the paper: DelSole and Tippett (2016) just apply the sign test (a special case of the binomial test with p=0.5), only that they visualize how the outcome develops from forecast case to case like a random walk. I would recommend to refer to the test simply as the sign test (which in fact dates back to 1710!).

Sect. 3.2 and Fig. 7: 1) Can you please explain how the bias correction is performed? Is this simply done for each gridcell individually? Do you just subtract the mean concentration bias (difference as a function of time of the year and lead time, averaged over 2011-2016/17), possibly with a correction that makes sure concentration values remain bound between 0 and 1? Or is quantile normalization involved? 2) Related to

the bias correction, I would find it very useful if the forecast errors could be compared against a climatological benchmark forecast. The latter could be based simply on the same period (2011-2016/17), or on the preceding decade (to make it more "operational"). I would expect that the uncalibrated forecasts are worse than climatology for most lead times (except the first one or two months?), but the calibrated might beat the climatology for a few months? In the summary section you say very clearly that you are "yet to demonstrate the benefit of interannual varying data on bias-corrected forecast scores", but I think it would be rather easy and revealing to add a climatological reference (even if it reveals clear limitations of current sea-ice forecast skills).

Fig. 8, top: Can you provide an explanation why the SIV in the ORAs converge from May to September, so that the large SIT difference in spring is completely "forgotten", whereas the coupled forecasts maintain much of the initial offset? Is there some fundamental reason why the forced (vs. coupled) atmosphere would cause such a differenc, or can it be linked to the continued assimilation of SIC (or ocean variables)?

Eq. 2: The way the melt energy tendency is defined, is seems to be really just the derivative of (area-averaged) SIT (times a constant factor), right? Also, maybe it's better to use partial d's to make clear that these are not material (Lagrangian) derivatives. Related, you could also mention that changes in SIT through divergence as well as advection are included, implying that the "melt energy tendeny" can in principle also change through dynamics. I understand that, by averaging over a large area (almost the whole Arctic), most of any dynamical effects would be compensating each other, but being clear about this would be good.

L314-316 & Fig. 9: The plot caption reveals that for the forecasts you look only at the first-month MET, but you do not mention/explain/motivate this in the text. Further, do I understand correctly that, by considering just the first month of the respective forecasts instead of a "closed" seasonal cycle, the annual integral of MET is not expected to be zero (while it should be zero for the ORAs)? In fact it looks a bit like it's rather negative (average build-up of sea-ice volume), can you confirm this?

Fig. 10 and corresponding text: I am wondering to what extent turbulent fluxes (in particular sensible) could also play a role, for example, with stronger downward spring/summer sensible heat fluxes in the forced ORAs compared to the coupled forecasts (acknowledging that there might not be a corresponding observational dataset to compare against). Too high near-surface temperatures that could generate too strong downward sensible heat flux would be consistent with a positive downwelling longwave bias in ERA-I, even if clouds also seem to play a role there. If differences in turbulent fluxes are too small to be important, please mention that.

L351-352: "Significant cold biases are present in forecasts for most of the start months and lead months" - Is this also true over Arctic sea ice in winter? If so, how can it be reconciled with Batrak and Müller (2019) "On the warm bias in atmospheric reanalyses induced by the missing snow over Arctic sea-ice"? I thought that the surface coupling is similar in the system studied here?

Fig. 12: I was a few times slightly confused when looking at this figure, intuitively thinking that the lower panels show differences between FC-SIT and FC-REF that could be directly combined with the biases shown in the upper panels. But the lower panels show the differences in mean absolute error, which is alright but easily misleading. I suggest to use a different colourbar for the lower panels so that the different flavours of "temperature" (signed vs. unsigned) is more intuitively reflected.

########## # Technical corrections

L25: last -> lasts

L80: "as cross-check variables evaluation" - I recommend to reformulate.

L91: These -> This

L168: "differ on" -> "differ in" / "differ regarding"

L208: "gradients on" -> "gradient in the" or "gradients of the"

L212: "end of melt season" -> add "the"

L217: "reduced up to" -> "reduced by up to"

L227: "indicates" -> "indicate"

L228: "at marginal seas" -> "in the marginal seas"

L232: "on an average" -> "on average"

L232-233: "in ORA-SIT analysis" -> add "the"

L265: "that to be" -> add "are"

L288: "is smaller" -> "are smaller"

There are a few more such tiny things, please check carefully!
* * *
[Figure]

Balan-Sarojini and co-authors present a study examining the impact of sea ice thickness (SIT) assimilation on seasonal forecasts of the northern hemisphere sea ice cover. In its approach and scope the study covers new ground; several of the key findings are substantive and represent a significant advance in our understanding of sea ice predictability and performance of seasonal-scale forecasts. The authors make good use of newly available, state-of-the-art ice thickness fields and strike a nice balance between more fundamental questions of prediction system performance, and applied questions related to improving forecast skills of Arctic sea ice models.

The paper is well structured and makes good use of figures to illustrate key points. The scientific approach is well described and appropriate for the problem at hand. The first half of the paper (up to and including Section 3.2, Fig. 7) is particularly compelling

and self-contained. The latter part of the manuscript, while interesting, is less compelling with some of the writing lacking clarity and the paper losing focus with respect to the goals laid out in the introduction and implicit in the title. If this part of the paper is retained in full, tightening the text and improving readability of sections 3.3-3.5 in particular would make the paper more accessible.

At the same time, a few aspects of the paper can be improved or require further thought, as outlined below.

First, given the central role the SMOS/Cryosat-2 data set plays in this study, one would like to see some discussion of how errors and uncertainties in the ice thickness data set may have impacted forecast skill and in particular some of the regional patterns observed in the thickness-assimilation runs. As shown in Ricker et al. (2017) uncertainties due to the fundamentally different retrieval approaches for SMOS and Cryosat-2, and to a lesser extent the optimal interpolation and data merging schemes, vary significantly by region. For example, over the Canadian Basin with mostly thick, multiyear ice the data product is dominated by bias/errors in Cryosat-2 data whereas in the Bering or Okhotsk Sea thinner ice weights errors towards those associated with SMOS thickness retrievals. It would be important to establish whether differences in thickness-field uncertainties have any impact on model performance and regional or temporal contrasts in forecast bias. This is also relevant for the integrated analyses of parameters such as the Integrated Ice Edge Error or ice volume at the pan-Arctic scale which may gloss over important regional contrasts in model performance.

Second, the paper lacks detail on the representation of ice thickness and key ice growth, melt and deformation processes in the LIM2 prognostic thermodynamic-dynamic sea ice model used in this study. It would be important to provide more detail, in particular as to whether any of the parameterizations that are part of the Fichefet & Morales Maqueda (1997) – FMM97 – model have been updated or changed. Of potential concern in FMM97 – based on description in their original paper – would be the limited representation of surface melt processes and their impact on ice albedo as

well as physically unrealistic representation of internal ice melt (with internal "storage" of solar heat up to a 50% threshold). These shortcomings, which may have been addressed in updates but if so the paper needs to make this explicit, do not necessarily limit applicability of the model in the context of seasonal ice forecasts. However, they are problematic in diagnosing some of the linkages between surface forcing, energy storage and the seasonal ice cycle explored in Section 3.3, since FMM97 in its original form may be ill suited to examine in particular the spring-summer-fall transitions in terms of the surface radiation balance or rates of ice thinning and decay.

Given these potential concerns, it would be instructive in Section 3.3 to examine the proportion of up/downwelling shortwave fluxes (or albedo, for that matter) to get a better perspective on the sensitivity of sea ice as represented in FMM97 to variations in downwelling shortwave energy. Such a detailed analysis may well be beyond the scope of the present paper. If so, this may be an argument to remove the latter parts of the paper as the basis for a separate, more detailed study. The first part of the paper (up to Section 3.3) is substantive enough and fully in line with the title of the paper.

Third, starting with the discussion of sea ice volume at the pan-Arctic or northern hemisphere scale the paper began to veer off-course a bit in terms of the goals laid out in the introduction. While total ice volume is a great integrator and a relevant variable in a global context, I was not able to tell whether the authors were assuming that it can also serve as an integrated measure of model performance in terms of ice concentration/extent and ice thickness. Given the regional contrasts in model performance apparent in the early figures of the paper the wholesale discussion of ice volume is somewhat problematic. For example, the interpretation of the seasonal ice volume cycle in terms of a single "freezing rate" (p. 17, top paragraph) is too simplistic since increases in ice volume during fall and winter occur through a combination of ice deformation and ice growth inside the ice pack as well as advance of the ice edge in marginal seas. Without an in-depth analysis some of the earlier figures and a solid understanding of how well the sea ice model is capturing the relevant processes, Figures such

as Fig. 8, don't add that much to the paper and could be relegated to supplemental materials or cut completely.

Finally, just a few minor points: - Comparing bias in ice thickness (Fig. 1) with bias in ice concentration (Fig. 3) it's striking that regions with near-zero bias in thickness (e.g. East Siberian Sea, Chukchi Sea in November) show up as having significant bias in ice concentration; moreover despite substantial contrasts in thickness biases between reference and ice thickness runs (Fig. 1c&d) the biases in ice concentration are near indistinguishable (Fig. 3 g&h). How can this be explained? - In regards to July ORA-SIT biases in ice concentration, it was striking to see much larger bias in the ORA-SIT than in the reference runs. Why would the simulations that performed (understandably) so much better in replicating ice thickness in March through assimilation of ice thickness data perform much worse in replicating ice concentration in July? Note that this finding also seems to contradict your statement in l. 185 that "The non-availability of the observations for the melt season in a way provides an opportunity to test the predictability of winter SIT from summer initial conditions." - You discuss your findings in terms of Arctic ice concentration and thickness but your figures include regions outside of the Arctic proper (such as the Okhotsk Sea). Please clarify whether both model output and assimilated data cover the entire northern hemisphere sea ice or a subset of that data. This is relevant in particular for figures like Fig. 5 which references "nh" in the figure label (for northern hemisphere?) but refers to Arctic sea ice area in the caption.

Minor comments & corrections

l. 2/3: change to "in its early stage"

l. 20 "near-surface temperature forecasts of early freezing season initialized in May": This phrase is confusing and not entirely clear, please revise to clarify what specifically is forecast with respect to "freezing season".

l. 25: change to "lasts into autumn"

l. 80: "it is relevant as cross-check variables evaluation" – not entirely clear what's referenced here – should it be "they are relevant because they allow for cross-checking between variables"? Please clarify.

l. 81: "SIT verification is also conducted as a sanity check of the nudging approach" – You lost me at "sanity check" – what exactly are you doing here? Please explain.

l. 91: change to "The Level-3"

l. 145: "LIM2 has a single sea ice category to represent sub-grid scale ice thickness distribution" – this needs further clarification. To calculate an effective conductive heat flux through the ice Fichefet and Morales Maqueda (1997) assumed a uniform thickness distribution bounded by zero and twice the average thickness. This parameterization was only applied in calculating heat fluxes through ice and lateral melt rate but did not enter into any of the ice dynamics components of the model. Given that ice thickness initialization is central to this manuscript, a clearer description of what exactly was implemented is needed.

l. 168: change to "differ in"

l. 233: "These results clearly show..." – Some clarification is needed here, since I interpret Fig. 4 as indicating that through May (but not the entire melt season), the effects of SIT assimilation are evident, beyond that the reference run performs better through the end of melt. In linking SIC increments to SIT assimilation please also consider the points raised in the general comments above.

l.238: "(units are..." – This should be part of the figure legend or caption, and not be buried in the main text.

l. 245: change to "melt season forecasts are considerably reduced"

l. 251: The top labels of the figure panels are cut off and it's not clear that they're actually needed ("bias for sia in area nh" – would need to be explained; also: is nh Northern Hemisphere? If so, what is the difference between this data for northern

hemisphere and the Arctic sea ice area as indicated in the figure caption?); the color scale needs better labeling.

l. 265: insert "are" in "that are to be expected"

l. 268: Fig. 6 - This figure should be cleaned up a bit as well; there's no need for two top labels (the upper one is more descriptive anyways, but even that's not needed given the explanation in the caption); the color bar needs proper units. Fig 7: Same comments apply – the 1e12 and 1e11 squeezed right next to the figure panel label and disjunct from the axis label (with units of square meters) are less than ideal and need to be cleaned up.

l. 287: Fig. 8: It's not clear to me how an axis label of 10ˆ1 3 mˆ3 translates into 10ˆ12 mˆ3 as the figure caption claims. Why not put an axis label in kmˆ3?

l. 361, Figure 11: same comments as for Fig 6 apply

l. 369: correct spelling of "Atlantic"